# Case Order Effects in Legal Decision-Making

**DOI:** 10.3390/bs15091250

**Published:** 2025-09-14

**Authors:** Paul Troop, David Lagnado

**Affiliations:** 1Psychology Department, University College London, London WC1E 6BT, UK; d.lagnado@ucl.ac.uk; 2Law School, The Open University, Milton Keynes MK7 6AA, UK

**Keywords:** decision-making, legal psychology, order effects, forensic psychology, psychology and law, moral decision-making, jury decision-making, trolley problems, neurolaw, experimental legal psychology

## Abstract

Case order effects, where decision-makers resolve dilemmas differently depending on the order in which cases are presented, are well established in the psychology of moral decision-making. Yet this type of order effect has rarely been studied in a legal context. Given the integral importance of consistency and precedent to the law, we sought to test for the existence of case order effects in legal decisions. Participants across five studies (total *n* = 1023) were given pairs of life-or-death legal cases to decide, consisting of one decision generally viewed positively in isolation, and one decision negatively viewed, with the order of presentation being varied (positive before negative vs. negative before positive). Studies included civil and criminal cases and individual and group decision-making. Results demonstrated that the case order effects previously seen in the moral context also held in the legal context. Order effects were asymmetric, with responses to one case remaining stable while responses to the other being labile, depending on the order presented. A particularly novel finding was of responses to labile cases becoming less, rather than more, similar to responses to preceding cases. Order effects can be readily triggered in the context of legal decision-making, suggesting legal precedent may be partially dependent on the order in which cases are determined. The asymmetric and previously undiscovered direction of these order effects is not consistent with existing consistency-type theories which predict effects to be symmetrical and more similar to previous cases and the findings are only partially consistent with salience-type theories.

## 1. Introduction

Case order effects, whereby an adjudicator makes different decisions depending on whether a pair of cases is presented in the order A–B or B–A, are deeply problematic as they suggest that arbitrary factors are influencing legal outcomes. In the absence of an onus to follow previous precedent, we assume that a fully rational adjudicator applies a consistent set of values to the facts of each case, meaning that the order in which cases are presented ought to be immaterial. Order effects may be particularly problematic in the legal context due to the practice of stare decisis, or ‘letting the decision stand’. Stare decisis is where subsequent courts follow the decisions of earlier courts, not because the earlier decisions are correct, but instead to provide certainty to litigants ([10]; [46]; [68]). Areas of the law established or developed by common law decisions may therefore be heavily influenced by the first case to be determined, implying that arbitrary influences on this first determination may be amplified in a legal context.

### 1.1. Case Order Effects in Moral Decision Making

Because case order effects would be so incompatible with the values of a legal system, they may have been assumed not to exist, which could explain the relative dearth of empirical research in this area. By contrast, in the domain of moral decision making, these order effects have been found to be widespread. A common paradigm is for participants to sequentially decide randomised pairs of analogous moral dilemmas, where a choice in one dilemma is generally approved of but disapproved of in the other. Often, one or both dilemmas are found to be determined differently depending on whether they are decided first or second. For example, many paradigms are based on hypothetical questions of runaway trams (or trolleys) that will run over and kill different numbers of people depending on the decision maker’s choice. Frequently, there are five people who will die if the trolley continues its trajectory, but only one person if the participant chooses to divert the trolley to another track ([71]). In ‘Switch’, a scenario where participants can divert the trolley from killing the five by pressing a switch, most participants approve of the intervention ([4]; [58]; [84]). By contrast, in ‘Fat Man’, a scenario where participants can prevent the killing of the five people by pushing a large man off a bridge into the path of the trolley to block it, but causing his death, most people disapprove ([4]; [58]). Most pertinent for our purposes is that Switch tends to be approved if assessed first, but disapproved of if assessed second, following Fat Man ([43]; [47]; [59]; [72]; [89]). While trolley dilemmas are presented as moral decisions, it is not difficult to see that these life-or-death decisions could readily attract criminal or tortious liability in a legal context. Yet while this research programme has triggered enormous empirical and theoretical work in the field of morality ([4]), there has historically been a relative dearth of research in the related legal field ([46]; [77]).

### 1.2. Empirical Evidence of General Order Effects in the Legal Context

Research on legal order effects generally has focused on the order of evidence presentation within a single case, looking for evidence of primacy or recency effects on the decision. Here, mixed results have been found, with some researchers finding primacy effects ([62]), most finding recency effects ([2]; [14]; [16]; [23]; [25]; [32]; [48]; [64]; [67]; [70]; [92]), and others finding both ([39]). In a more sophisticated manipulation, Pennington and Hastie found that evidence that was presented in chronological or story order was more persuasive than evidence presented in a random order ([63]; [64]).

The research into ‘anchoring’, where seemingly irrelevant information influences decisions, bears some similarity to the potential for the types of arbitrary effects of case order on legal decisions that we are interested in investigating, particularly where the anchor consists of a legal precedent ([10]). Anchoring has been widely demonstrated within decision-making generally ([5]; [13]; [29]; [30]; [56]; [60]; [82]). Thus, a participant asked to estimate a value will often be influenced to choose a higher value when they have been shown an arbitrarily generated value immediately prior to giving the estimate and vice versa. Given the extensive evidence for anchoring generally, it is unsurprising that anchoring has also been shown to occur in the legal domain. Thus, where an adjudicator has to select a value such as an amount of damages or a length of sentence, arbitrary anchors have been shown to be influential. For example, the sentence requested by a prosecutor, or the damages requested by a claimant, have been consistently shown to be correlated with sentence lengths and amounts awarded ([18]; [27]; [26]; [31]; [35]; [37]; [49]; [50]). Just as in psychology generally, anchors seem to be effective even when obviously randomly generated ([28]). While most of these anchors were in the form of arbitrary values presented before or during a case, Feldman et al. conducted an experiment related to a copyright infringement based on unauthorised sampling of 8% of another’s music ([31]). Anchors were provided in the form of previous precedents, where a 1% sample was held never to amount to a breach and another where a 50% sample was held to always amount to a breach. These precedent anchors had a significant influence on whether participants found the sample in the target case to be a breach of copyright.

The sort of arbitrary case order effects we are interested in should be distinguished from the effect of previous precedent on legal decision making because, as noted above, the principle of stare decisis obliges courts to follow precedent they may disagree with in order to provide predictability for litigants. Precedent should be particularly influential where it is from a higher court and is binding on the decision maker, sometimes referred to as ‘vertical’ precedent. By contrast, precedent is less influential where it is from a court at an equivalent level and only persuasive, so-called ‘horizontal’ precedent. Research has confirmed that judges make different decisions where vertical precedent exists compared to where it does not ([45]; [46]), and horizontal precedent has also been found to have an influence on judges ([40]; [68]; [77]).

### 1.3. Theoretical Explanations

If arbitrary case order effects are found to occur, it would be valuable to understand why, so that measures could be developed to minimise their occurrence. Different effects might point towards different solutions. Two main explanations have been discussed in the corresponding domain of moral decision making: (1) individual consistency, and (2) salience-type explanations.

Individual consistency-type explanations are subtly different from the sort of consistency that arises due to adherence to the doctrine of stare decisis. Stare decisis arises where judges are influenced to be consistent with other judges for the sake of certainty and predictability. This is sometimes termed ‘institutional’ consistency and clearly serves important values of the wider legal system ([45]). By contrast, ‘individual’ consistency-type explanations assume a decision maker is influenced by their own previous decisions and seeks to be consistent for their own reasons ([45]; [57]; [76]). One reason a decision maker might seek to be individually consistent is to avoid a perception of bias, carelessness, or ignorance ([24]). For instance, because dilemmas such as Switch and Fat Man are superficially similar with the same number of potential victims, there is something of an implication that the same principle, such as the consequentialist principle to ‘always save the most lives’, should be applied in both dilemmas. Apparent inconsistency can suggest that the decision maker has been influenced by impermissible factors, is incompetent, or was unaware of material information. Any of these factors may undermine the decision maker’s reputation, giving them an incentive to maintain an appearance of consistency by following a consistent principle in both cases ([24]; [66]; [79]; [81]; [88]). Many commentators have therefore suggested that the order effects seen in the moral decision-making sphere are due to a desire to maintain an appearance of individual consistency ([38]; [65]; [71]; [84]; [89]).

The other main type of explanation for these sorts of order effects is salience-type explanations. This is where the order effects are assumed to arise because the unique facts of one of the decisions draws the attention of the decision maker to factors that they would want to take into account when considering the other decision, but would have otherwise overlooked ([61]; [71]; [90]). For example, there is a moral dilemma called ‘Transplant’ in which a healthy individual could be murdered to transplant their organs to five other individuals requiring organs to survive ([59]). Though this would save five lives at the expense of one, most people intuitively recoil from such a choice, indicating that the consequentialist principle of saving the most lives is not always the appropriate solution ([65]; [72]; [89]). Thus, a target scenario may be determined differently if followed by the source scenario that makes particular information salient to the decision maker, but not if the target scenario is determined first or in isolation.

These two different types of explanations imply very different hypotheses. Consistency-type explanations imply that a decision taken second will tend to more closely resemble the decision that precedes it compared to when it is taken first or in isolation. For a pair of scenarios, responses to both would be expected to change to some extent, and responses to both would be expected to become more similar to the previous scenario when determined second. By contrast, salience-type theories would predict asymmetric changes, with responses to one scenario changing when determined second, but not necessarily the other scenario. And salience-type theories would imply order effects in either direction, with responses potentially becoming more similar or dissimilar to responses to the scenario that precedes it.

### 1.4. Our Research

Our aims in this research were therefore two-fold: first to ascertain whether apparently arbitrary case order effects could be replicated in a legal decision making context (both civil and criminal law), and secondly, to the extent that these order effects manifested themselves, to establish whether the two families of theories, consistency and salience theories, offered a plausible explanation for these order effects.

The approach we chose was to approximate the methodology used successfully in the moral decision-making field of trolley experiments, whereby pairs of superficially similar dilemmas provoking different intuitions were presented in forward or reverse order. However, whereas trolley experiments often rely on contrived or implausible scenarios, we sought to base our dilemmas on more plausible or actual legal scenarios, both civil and criminal in nature. Though the participants were overwhelmingly laypeople rather than professional legal adjudicators, the majority of the dilemmas tested were the type of legal decision that would in real life be taken by laypeople. In addition, while one study relied on a cohort of student participants from a relatively homogenous demographic, participants in the rest of our studies were more heterogeneous and representative of wider society.

Across both the civil and criminal contexts, we found statistically significant evidence for the existence of case order effects in pairs of analogous legal scenarios, thereby replicating the order effects commonly seen in the moral decision-making context. These order effects were in both directions: responses in some scenarios seen second became more similar to responses to scenarios seen first, whereas in other scenarios, responses to scenarios seen second became more dissimilar to responses to scenarios seen first. In addition, order effects were invariably asymmetrical, with responses to one of the scenarios remaining consistent (or ‘stable’) regardless of the position in which it was determined, whereas responses in the other scenario were changeable (or ‘labile’) depending on whether it was determined first or second. On the face of it, this pattern of results would seem to favour salience-type explanations rather than individual consistency-type explanations. However, different experimental manipulations designed to make the information assumed to trigger these order effects more salient to decision makers failed to induce the same effect, thereby raising questions about the adequacy of salience-type explanations.

## 2. Study 1

Study 1 was designed to see if the order effects identified in moral decision-making could be replicated in a legal decision-making context. The study followed a similar design to trolley-based moral decision-making but was implemented in a more realistic civil law context. The underlying legal principle for participants was whether it was lawful for an individual to commit suicide on hospital premises to facilitate the donation of their organs. In accordance with the law in England and Wales, it is no longer a criminal offence for an individual to commit suicide, but it is an offence for somebody to assist an individual attempt or commit suicide ([83]). In situations where the lawfulness of a proposed course of medical treatment is in question, the authorities may seek advance guidance from a civil court. For example, when it was legally uncertain whether it was lawful to remove treatment from a minimally conscious patient, a hospital trust treating him sought guidance from the High Court whether the discontinuance of medical treatment (save for purposes such as reducing pain) would be lawful ([7]). A desire for those intending to commit suicide to also become donors has been reported in the media and academic literature ([1]; [3]; [55]), the concept of using organs from individuals committing suicide has been discussed in the academic literature ([9]; [85]; [74]; [91]), and organ donation following suicide takes place in jurisdictions such as Belgium and the Netherlands where assisted suicide is legal ([8]; [33]; [86]; [87]; [93]). As such, it was considered a reasonably plausible, if unlikely, legal scenario, correspondingly with limited relevant precedent.

Following pre-testing, two scenarios were selected for the main experiment. In the first, the reason that the individual wished to commit suicide was because they were suffering from multiple sclerosis (‘MS’), a long-term incurable condition for which the symptoms can be addressed. When presented with the MS variant, most pretest participants agreed that the proposed course of action was acceptable (82% approval vs. 18% disapproval, *n* = 11). In the second scenario, the reason the individual wished to commit suicide was because they suffered from long-term depression. Most participants in the pretest instead objected to this (42% approval vs. 58% disapproval, *n* = 12).

Given previous research in the moral decision-making domain, we predicted order effects, depending on the order in which the scenarios were presented. Given the evidence of asymmetrical order effects, we also envisaged that one scenario may be more labile than the other, but we had no theoretical reason to predict which.

### 2.1. Method

#### 2.1.1. Sample/Participants

One hundred and ninety-nine participants recruited using the online survey platform Prolific completed the survey (124 females, 75 males; aged 40 to 65, *M* age 50.3, *SD* 7.4; 2.5% were students; 45% in full-time employment, 26% in part-time employment, 29% unemployed or other) and were paid GBP 0.50 for their time. The design was calculated to have an 80% power to detect an effect size of *d* = 0.40. Given the lack of prior research in this area, this effect size was chosen as being commensurate with a ‘medium’ effect size. Participants were selected on the basis of British nationality and residency in England and Wales. The study was conducted in accordance with approval obtained from UCL’s Research Ethics Committee (EP/2018/005). Informed consent was obtained from each individual in advance of participation by providing them with study information and a consent form to agree to.

#### 2.1.2. Design

We used a 2 × 2 mixed design in which all participants viewed both cases, the depression scenario and the MS scenario, sequentially, but in a randomised order. Given that all participants viewed both scenarios, this was a within-subject manipulation. Participants only viewed the scenario in one of two possible orders; hence, order was a between-subject manipulation.

#### 2.1.3. Materials

The materials were hosted on the online platform Gorilla Experiment Builder (www.gorilla.sc). The case materials consisted of two summaries of the facts of the case and the issue to be decided. Both summaries were essentially identical, other than the underlying cause of the individual’s condition. In each case, participants were told that there were two people in need of urgent organ transplants who would die if they did not receive a transplant soon, and an individual, hospitalised with the relevant condition, would be a suitable donor. Participants were told that, despite undergoing treatment, this individual wanted to die and also wanted to donate his organs. Participants were told that the individual wished to commit suicide in a hospital so that the hospital could receive his organs in a good condition.

In each case, participants were advised that while it was legal to commit suicide, it was not legal to assist an individual to commit suicide, and, for this reason, the individual and the hospital were making a joint application to the court to seek guidance as to whether it was lawful for the individual to commit suicide on the hospital premises.

Once participants had read the information in each scenario, they were asked to imagine that they were the judge and to make a decision on the acceptability of the particular application. Participants were first asked whether they would or would not grant the application, then asked secondly to indicate on a percentage scale from 0 to 100 the acceptability of the application, and then thirdly to explain the reasoning behind their decision.

#### 2.1.4. Measures

As noted above, participants were asked to indicate three measures. For reasons of external validity, participants were asked to give a ruling on whether they would grant or oppose the application, and their reasons for this, both of which would be matters that would ordinarily be expected of a real-world adjudicator. In addition, to provide more insight into participants’ cognition, as well as to be consistent with the measures typically taken in trolley-type moral decision-making research, participants were also asked to indicate their view of the acceptability of the application on a 100-point percentage scale, where the extremes of the scale represented completely unacceptable and completely acceptable.

#### 2.1.5. Procedure

On referral from the Prolific platform, participants were first provided with the study information form. They were then asked to complete the consent form, comprising a number of statements to which an affirmative answer was required in order to participate. Users were identified anonymously to enable subsequent matching of demographic data without compromising participants’ personal information.

Participants were randomly assigned to one of the two conditions, such that half of the participants viewed the MS scenario followed by the depression scenario, and half viewed the depression scenario followed by the MS scenario. Participants were required to give responses to the first scenario viewed before they could complete the second scenario. After reviewing the first scenario, participants were asked to indicate whether they would grant or oppose the application, to indicate how acceptable they found the application on a 100-point % scale from 0 to 100 and were asked to explain the reasons behind their decision. Once they had completed the first scenario, they were then presented with the second scenario and asked to complete the same measures.

After completing the survey, participants were thanked for their participation and referred back to the Prolific survey platform to confirm their participation.

### 2.2. Results

In accordance with pre-testing, participants found the depression application less acceptable (*M* = 32.8%, *SD* = 34.0, *n* = 101) than the MS application (*M* = 57.9%, *SD* = 33.3, *n* = 98) when they were considered first. A two-sample *t*-test confirmed that this difference was statistically significant at the 0.05 level (*t*(197) = 5.27, *p* < 0.001, 95% *CI* [15.73, 34.55], Cohen’s *D* = 0.75). Correspondingly, participants were less minded to grant the depression application (80:21 refuse:grant) than the MS application (37:61 refuse:grant) when these were considered first. A Fisher’s exact text confirmed that this difference was statistically significant (*p* < 0.001, *OR* 6.21, 95% *CI* [3.20, 12.45]).

As predicted, there was an order effect, and this was asymmetrical, with the depression scenario being stable, whereas the MS scenario was labile, see Figure 1. Thus, the depression scenario remained overall unacceptable whether seen in either first or following the MS scenario, even becoming slightly more unacceptable when seen in second (32.81% vs. 27.16% acceptability). However, this small difference was not significant (*t*(197) = 1.25, *p* = 0.21, 95% *CI* [−3.27, 14.56], Cohen’s *D* = 0.18). Correspondingly, participants generally refused the depression application, whether it was seen first (80:21 refuse:grant) and were slightly less likely to grant it when it was reviewed after the MS scenario (89:9 refuse:grant). A Fisher’s exact test indicated that this difference was significant (*p* = 0.029, OR 0.39, 95% *CI* [0.15, 0.95]).

By contrast, responses to the MS scenario changed much more drastically, flipping from overall approval to overall disapproval, as shown in Figure 1. The MS scenario received a mean acceptability well over 50% when seen in isolation (*M* = 57.9%, *SD* = 33.3, *n* = 98), but this dropped to well below 50% when reviewed after the depression scenario (*M* = 43.4%, *SD* = 35.1, *n* = 101). A two-sample *t*-test confirmed that this difference was statistically significant (*t*(197) = 3.00, *p* = 0.003, 95% *CI* [5.00, 24.13], Cohen’s *D* = 0.43). Unsurprisingly, decisions to approve or reject the application changed similarly, such that most participants granted the MS application when seen in isolation (37:61 refuse:grant) but refused it when it was reviewed after the depression scenario (62:39 refuse: grant). A Fisher’s exact test was consistent with this (*p* = 0.001, *OR* = 0.38, 95% *CI* [0.21, 0.70]).

### 2.3. Discussion

The main finding of this study is that it is possible to replicate the types of order effects seen in moral decision making in the domain of legal decision making. In particular, when presented first, the MS scenario was found to be generally acceptable to participants on both the continuous variable that measured acceptability and the categorical variable that measured whether participants would grant or refuse the application. But when the MS scenario was considered after participants had determined the depression scenario, it was generally found to be unacceptable, and the application tended to be refused. In addition, an asymmetric order effect was found, which is a pattern similar to that seen in the moral decision-making context. In contrast to the lability of responses to the MS scenario, responses to the depression scenario were much more stable. When seen in isolation, the depression scenario was both generally disapproved of and, correspondingly, the application tended to be refused. And responses to the depression scenario hardly differed when considered following the MS scenario, with participants still finding it generally unacceptable and refusing the application. While there was some evidence of a slight difference in responses to the depression scenario when it was considered after the MS scenario, in that participants found it slightly less acceptable, the difference was only statistically significant on the categorical measure.

The findings are somewhat consistent with both categories of explanation. The finding that responses to the MS scenario become more similar to responses to the depression scenario when that precedes it fits with a desire for individual consistency, though the stability of responses to the depression scenario requires further explanation. One possibility is that the range of reasonable responses or the ‘zone of reasonableness’ is narrower for some scenarios ([66]). For example, the trolley problem ‘Transplant’ mentioned above is structurally similar to other dilemmas in that a surgeon could kill one healthy person to save five others requiring organ donations ([59]). Unsurprisingly, few participants ever find this an acceptable course of action, yet the Transplant scenario seems to cause order effects in other scenarios considered following it ([65]; [89]). The findings are also somewhat compatible with salience-type theories, which suggest that one scenario may highlight issues that are salient to the decision, but that the decision maker might otherwise overlook. For instance, the depression scenario might have been effective in highlighting issues such as the possibility of treating or alleviating the symptoms of the condition, or ‘slippery slope’-type arguments that permitting those committing suicide to donate organs in some situations might lead to people being permitted or implicitly encouraged to take this course of action in more trivial circumstances.

While the materials and nature of the decisions were more legally plausible than the parallel field of trolley decision making, this type of civil law decision would always be made by an experienced High Court judge, and the participants here were laypeople. This is a practice common in the parallel field of moral psychology, where lay participants are often asked to assume the role of professionals such as medical professionals, for example, in the Transplant scenario. The broader experience of a High Court judge could well lead to different patterns of decision making as we know that professional judges draw more heavily on experience and precedent than lay decision makers ([69]; [78]). For instance, to the extent that the order effects revealed by Study 1 are the result of participants overlooking certain salient aspects of the decision, an experienced judge may be less likely to overlook these aspects. One means of examining this would be to repeat the experiments with professional judges, but given the obvious resource implications, it makes sense to establish the clear existence of the phenomenon with lay decision makers first. To this end, we next sought to determine the existence of order effects in the context of a criminal law decision that would ordinarily be taken by laypeople.

## 3. Study 2

Study 2 sought to replicate the findings of case order effects in the context of a criminal decision that would usually be undertaken by lay people. We again sought to use the standard paradigm of presenting two cases sequentially, where one case was generally assessed favourably in isolation, while the other case was generally assessed negatively in isolation. However, because penal decisions are primarily binary ones about the guilt or innocence of an accused rather than decisions that have consequences for lives lost or saved, the context was necessarily slightly different. We therefore asked participants to rule whether an accused was guilty or not guilty after making a decision that had led to loss of life. The inspiration for the scenarios was a relatively historical but notorious common-law case of [6] ([6]). In the real-life case, two sailors who had been shipwrecked survived by killing and eating a cabin boy. They were subsequently prosecuted for murder. The sailors argued the defence of necessity, also known as duress of circumstances. In that case, judges ruled that the defence of necessity was not available for murder, and the sailors were convicted.

As with our first study, we created two versions of this scenario with contrasting levels of acceptability. The less acceptable version (‘apprentice’) was very similar to the facts in the original case of *R v Dudley v Stephens*: three sailors were shipwrecked and surviving on a liferaft. In this version, two of the senior members who were starving, the captain and the first mate, unilaterally attacked and killed the more junior apprentice, and consumed him. In pre-testing, this scenario was found to be generally objectionable with a relatively low mean acceptability of 39.4% on a 100% scale from completely unacceptable to completely acceptable and, correspondingly, verdicts were generally guilty (guilty:not guilty = 6:2). By contrast, in the more acceptable version (‘captain’), the three crew members discussed how to proceed and agreed to draw lots to decide who would be sacrificed. The captain happened to be the one who drew the short straw, and he allowed the mate to kill him before the two survivors consumed him. In pre-testing, this was seen as more acceptable (76.2%) and correspondingly verdicts tended to be not guilty (guilty:not guilty = 3:9). As with the first study, the facts were not in dispute, so the participants were not required to engage in fact-finding, only decision-making. As before, each participant considered both scenarios in a sequential order, with the order or presentation being randomised between participants.

Given previous research in the moral decision-making context and our findings from Study 1, we predicted a case order effect. Due to previous findings, we also foresaw as a possibility that responses would be asymmetrical, such that responses in one scenario would be more labile with responses in the other scenario would be more stable. However, because these scenarios were novel, we had no particular theoretical basis to predict which scenario would be labile and which would be stable.

### 3.1. Method

#### 3.1.1. Sample/Participants

Two hundred and ten participants recruited using the online survey platform Prolific completed the survey (129 female, 71 male; aged 18 to 71, *M* = 34.7, *SD* = 12.1; 17.5% were students; 50% in full-time employment, 21.5% in part-time employment; 28.5% unemployed or other) and were paid GBP 0.70 for their time. The sample was calculated to have an 80% power to detect a minimum effect size of *d* = 0.39. Participants were selected on the basis of British nationality and residency in England and Wales. The study was conducted in accordance with approval obtained from UCL’s Research Ethics Committee (EP/2018/005). Informed consent was obtained from each individual in advance of participation by providing them with study information and a consent form to agree to.

#### 3.1.2. Design

We used a 2 × 2 mixed design in which all participants viewed both cases, the apprentice scenario and the captain scenario, sequentially, and in a randomised order. As with Study 1, as all participants viewed both scenarios, the scenario was a within-subject manipulation. By contrast, participants only viewed the scenarios in one of two possible orders, so order was a between-subject manipulation.

#### 3.1.3. Materials

Participants were advised that they would be put in the place of a juror in a Crown Court and asked to consider two hypothetical cases where the accused was relying on the defence of duress of circumstances, and that they would be asked to determine whether the accused should be convicted or acquitted. It was explained to participants that the defence of duress of circumstances is where an accused commits what would otherwise be an offence in order to save a life or prevent serious harm to somebody.

In both scenarios, it was explained that a small 3-man crew were employed to transport a new yacht to Australia and that part-way through the voyage and when a long distance from land, the yacht was fatally damaged in a storm and rapidly sank. The crew escaped to the lifeboat, but without communication equipment. Participants were told in both cases that over a period of 2 weeks, the crew exhausted the lifeboat’s rations, they were unable to catch fish or seabirds and only collected a tiny amount of water. After a further week with no provisions, the situation became desperate. The crew had seen no other ships in the 3 weeks, estimated that they were around 1000 miles from land, and that there seemed no immediate prospect of rescue. Thereafter, the scenarios diverged. In the apprentice scenario, participants were told that the captain and mate secretly discussed the situation and decided that they might survive for a little longer if they killed and ate the apprentice, who they felt would be likely to die soonest. Participants were advised that after the apprentice fell asleep, the mate held him down while the captain killed him with a knife. Over the subsequent days, the captain and the mate consumed the apprentice. By contrast, in the captain scenario, participants were advised that the 3 crew members discussed the situation together and decided to draw lots to decide who would be sacrificed. The captain drew the short straw and allowed the mate and apprentice to kill him with a knife. Participants were told that over the subsequent days, the mate and apprentice consumed the captain. Participants were then told in both scenarios that 6 days later, a ship was seen, and the remaining sailors used their final distress flare successfully to attract its attention, leading to their rescue. The rescuers witnessed the remains of the sailor who had been killed.

In both scenarios, participants were given further details of the legal issues, in particular, the defence of duress of circumstances. This included the information that (1) duress of circumstances is where the accused committed the offence because they reasonably believed that they would die or be seriously injured, and (2) a sober person of reasonable firmness, sharing the characteristics of the accused, would have acted in the same way. Because the scenario was this time a criminal case, it was explained that the burden of proof was on the prosecution. Thus, participants were advised that if they were sure that one or both of those statements was untrue, they should find the accused guilty, and if they thought that both of those statements were or may be true, they should find the accused not guilty.

Participants were then asked how they found the accused, guilty or not guilty, to indicate on a scale from 0 to 100 how reasonable it was for the accused to argue the defence of duress of circumstances, and they were also asked to explain their decision.

#### 3.1.4. Measures

As with the previous experiment, participants were asked to respond to three measures. The first measure was simply a verdict on the accused, as would be expected in a criminal trial. This was a binary choice between guilty or not guilty (for both accused). Similar to the previous experiment, in order to glean a more nuanced understanding of participants’ views as well as for consistency with previous moral psychology trolley-type research, participants were also asked to give a view of the perceived reasonableness of the accused arguing duress of circumstances. This was a 100-point scale from 0 (completely unreasonable) to 100 (completely reasonable) with 1-unit gradations. Finally, as with the earlier experiment, participants were also asked to explain their decision using an open text response field. While this survey was based on a criminal jury trial in which the jury would not be expected to give public reasons, individual jurors in a criminal trial would be expected to give their reasons to the other jurors as part of the group deliberations, so the request for an explanation seemed appropriate.

After participants had reviewed the scenarios, they were also asked some further questions. One question was whether they thought that their response to the first scenario affected their response to the second scenario, with participants permitted to select from a ternary of ‘yes’, ‘no’, and ‘don’t know’. Participants were then asked to indicate a binary response to which statement best reflected their view: ‘similar legal cases should be treated the same’ or ‘each legal case should be decided on its own merits’. Participants were also asked as an attention check to indicate what they were asked to decide given a choice of three statements of which they could choose as many options as they wished: ‘whether the accused should be acquitted or found guilty’, ‘whether the accused should be allowed to argue defence of circumstances’, and ‘whether the case should be referred to the Court of Appeal’. Participants were finally asked whether they found the explanation of the law easy or difficult to understand and to explain if they had any relevant legal experience or knowledge.

#### 3.1.5. Procedure

As noted above, participants were recruited online and participated in the survey using the online platform Qualtrics in a place of their choosing, using their own device. On initial referral from the Prolific platform, participants were first provided with the study information form. They were then asked to complete the consent form comprising a number of statements to which an affirmative answer was required in order to participate in the survey. An anonymous user identification was collected to enable subsequent matching of demographic data without compromising the participants’ anonymity.

Participants were randomly assigned to one of the two conditions such that half of the participants viewed the apprentice scenario followed by the captain scenario, and half the participants viewed the captain scenario followed by the apprentice scenario. After reviewing the first scenario that they were allocated to, they were first asked whether they would find the accused guilty or not guilty, secondly how reasonable they found the defence on a scale of 0 to 100, and thirdly to explain their decision. Once they had reviewed and responded to the measures on the first scenario, they were then presented with the second scenario. After completing the responses to the scenarios, they were posed the additional measures referred to previously.

After completing the survey, participants were thanked for their participation and referred back to the Prolific survey platform to confirm their participation. Once both platforms had confirmed the participant’s successful completion of the survey, their remuneration was authorised.

### 3.2. Results

Most participants (79%) found the explanation of the law easy to understand, with a small proportion (21%) finding it difficult. For the attention check, all participants indicated that the issues in the case were whether the accused should be acquitted or found guilty or whether the accused should be allowed to argue defence of circumstances or both, with no participants indicating the erroneous option of whether the case should be referred to the Court of Appeal. A small proportion (5%) had some modest legal familiarity, such as having studied a law degree, but only a handful (1%) had heard of the case of *R v Dudley v Stephens*. As such, the decision was taken not to exclude any participants.

Responses to the scenarios considered in the first position were consistent with responses given in pre-testing. Participants were much more likely to find the accused guilty in the apprentice scenario (73:30 guilty:not guilty), and this pattern was reversed in the captain scenario (37:62 guilty:not guilty). A Fisher’s exact test confirmed that this pattern was statistically significant (*p* < 0.001, *OR* = 4.05, 95% *CI* [2.17, 7.67]). Similarly, the mean reasonableness assessment in the apprentice scenario was lower at 51.5/100 than that in the captain scenario of 69.3/100. This difference was statistically significant (*t*(200) = 4.36, 95% *CI* = [9.78, 25.9], Cohen’s *D* = 0.61, *p* < 0.001), see Figure 2.

However, while an asymmetric order effect of the scenarios was found, with the apprentice scenario being labile while the captain scenario being stable, the most interesting finding was that the order effect was in the opposite direction to that generally found, with participants more likely to convict and to view the accused as less reasonable in the apprentice scenario when they considered it after the (more acceptable) captain scenario: see Figure 2. Responses in the captain scenario remained consistent and fairly sympathetic to the accused, regardless of whether the scenario was considered first or second. Mean reasonableness assessments remained high in the captain scenario when viewed first (69.3/100) and only dropped slightly when viewed after the apprentice scenario (64.4/100). This difference was not statistically significant (*t*(200) = 1.26, 95% *CI* = [−2.77, 12.64], Cohen’s *D* = 0.18, *p* = 0.21). Similarly, the proportion of participants giving a guilty verdict in the captain scenario increased slightly after viewing the apprentice scenario (from 37:62 to 49:54 guilty:not guilty), but this difference did not reach statistical significance according to a Fisher’s exact test (*p* = 0.16, *OR* = 0.66, 95% *CI* = [0.36, 1.20]). As noted, contrary to usual patterns, responses in the apprentice scenario were labile, but became less similar to the captain scenario that preceded it. Thus, assessments of reasonableness decreased in the apprentice scenario from when it was considered first (51.5/100) to when it was considered after the more acceptable captain scenario (36.2/100). This difference was statistically significant (*t*(200) = 3.62, 95% *CI* = [6.97, 23.62], Cohen’s *D* = 0.51, *p* < 0.001). In line with this finding, guilty verdicts increased from 73:30 (guilty:not guilty) to 81:18, a difference that was borderline statistically significant according to a Fisher’s exact test (*p* = 0.07, *OR* = 0.54, 95% *CI* = [0.26, 1.10]).

In terms of the principles that participants were willing to endorse, most participants overall (88%) thought that each legal case should be considered on its own merits, and only a minority (12%) endorsed the principle that similar legal cases should be treated the same. Despite this, a majority (63%) of participants in fact gave a consistent verdict across the two scenarios, with only a minority (37%) giving inconsistent verdicts. Notably, participants who gave consistent verdicts were more likely to endorse consistency as a principle (16%) compared to those who gave inconsistent verdicts (5%). This difference was statistically significant based on a Fisher’s exact test (*p* = 0.03, *OR* = 0.29, 95% *CI* = [0.07, 0.92]), consistent with either participants’ values influencing their verdicts, or participants’ verdicts influencing the values they were prepared to endorse publicly.

Concerning participants’ subjective insight into whether they felt they had been influenced by the previous case, most (61%) said that they did not feel that they had been influenced. A corresponding minority (39%) felt that they had been influenced. Those participants who felt that they were influenced by the prior case were more likely to give a different verdict between scenarios (45%) than those who felt that they were not influenced who were less likely to give a different verdict between the scenarios (31%). This difference was borderline statistically significant at the 0.05 level according to a Fisher’s exact test (*p* = 0.09, *OR* = 1.76, 95% *CI* = [0.92, 3.38]). Given the novel finding that the order influence appeared to be that the captain scenario (in which participants were more likely to acquit) made participants subsequently see the apprentice scenario more likely to convict, the evidence seemed somewhat consistent with participants’ views.

### 3.3. Discussion

The research paradigm used in Study 2 seemed to confirm the existence of further order effects in paired dilemmas in the new legal context of the criminal law. Consistent with previous findings, the order effect was asymmetric, with one dilemma seemingly being influenced by the other, but not vice versa. The captain scenario, where the starving sailors killed and ate the most senior sailor on board following a fair selection process, was generally found more acceptable to participants. Thus, participants rated the sailors’ behaviour as more reasonable, leading them to prefer acquittal. This pattern remained fairly stable regardless of whether the captain scenario was considered first (and thus in isolation) or whether it was considered after the generally less acceptable apprentice scenario. By contrast, participants in the apprentice scenario, where the starving sailors killed and ate the most junior sailor on board following an unfair selection process, tended to find this much less acceptable. Correspondingly, participants rated the sailors’ behaviour as less reasonable and preferred to convict. However, the pattern of responses was much more labile in the apprentice scenario, with participants finding the sailors’ behaviour more reasonable and convicting more when they considered this scenario in isolation, compared to when they considered it after the captain scenario.

The unexpected direction of the order effect was particularly novel. Previously reported results of order effects in paired dilemmas from the moral decision-making field invariably indicated order effects of responses in similar dilemmas becoming more similar to one another. However, in this study, we found the influence had the opposite effect: responses in the labile dilemma became more dissimilar to responses in the dilemma that preceded it. While this is an initial finding, it is potentially quite consequential, in particular for explaining order effects theoretically. In particular, this finding seems inconsistent with individual consistency-type explanations because the responses are superficially less consistent with the responses that precede them. As such, while it seems likely that there will be some situations where participants try to appear consistent when considering similar dilemmas, particularly where the inconsistency implies that they have taken impermissible factors into account when reaching their decision (e.g., [57]; [76]), it does not seem likely that individual consistency is the most important influence in these studies. By contrast, the findings do seem to accord with salience-type theories that suggest that order effects may be caused by earlier scenarios drawing participants’ attention to aspects of the situations that they knew of but had not realised the salience of. Obviously, salient information could affect responses in either direction. Nonetheless, many questions remain unanswered, such as precisely what about the different information presented in the captain scenario might be salient. One speculation is that the behaviour of the sailors in the apprentice scenario might appear quite reasonable on a relatively superficial preliminary examination, but that the presentation of the captain scenario might cause participants to focus on salient information that they had not previously considered, such as that a fairer way of selecting the individual to be cannibalised was possible that did not lead to the death of the most vulnerable sailor.

An issue with the previous study was that participants were laypeople when the type of legal problem posed was one that would invariably be taken by a relatively senior judge. The topic of the present study addressed this fact to some extent in that the type of decision taken was one that would ordinarily be taken by laypeople, and the participants chosen would have been eligible, in principle, to act as jurors. There were, nonetheless, issues with the validity of the experiment, most prominently the fact that there was no group deliberation dimension to the participants’ task and that ordinarily jurors are not expected to give reasons for their group decision. Nonetheless, as real-life jurors would necessarily give reasons for their thinking to other jurors during the deliberation phase, the requirement to give reasons was consistent with this, even if there was no equivalent of deliberation. Nonetheless, the paradigm was useful for revealing the phenomenon, particularly given the considerable complexity and cost of carrying out jury research.

## 4. Study 3

The finding of an asymmetric order effect that caused participant responses to become more dissimilar to previous responses suggested two further avenues to explore in Study 3. The first was to replicate the effect to reduce the probability that the findings were due to stochastic factors. The second was to seek to shed further light on the potential causes of the effect. Because the effect was more consistent with the theory that a preceding scenario drawing participants’ attention to salient matters that would otherwise have been overlooked, it appeared reasonable to test whether this effect could be elicited using an alternative method. Thus, in addition to a standard replication, an additional condition was added, which was to disclose the information assumed to be driving the effect explicitly to participants prior to them determining the apprentice scenario.

In order to do this, we adopted a different experimental design. As before, all participants determined the apprentice scenario. In a first, control condition, participants determined only the apprentice scenario. In a second, replication condition, participants determined the apprentice scenario after determining the captain scenario. In a new third, test condition, participants determined a modified version of the apprentice scenario. This scenario was changed so as to transmit the information inferred to be causing the order effects, namely, the possibility of a fairer means of selection that was less likely to result in exploitation of the most vulnerable member of the crew. This modification was achieved by introducing closing submissions on behalf of both parties. In these submissions, the putative prosecutor explicitly argued that the behaviour of the accused was not reasonable because there were fairer means of selecting who to sacrifice, such as by randomly drawing lots. To avoid the potential confounding effects of persuasion, the prosecutor’s submissions were neutral and descriptive, and the defence submissions were also concise and banal.

We expected that the effect seen in Study 2 would be replicated and, if salience theories were correct, that a similar effect would be seen in the new test condition, whereby participants would find the acceptability of the accused’s behaviour in this condition to be less acceptable than in the control condition.

### 4.1. Method

#### 4.1.1. Sample/Participants

Three hundred participants were recruited using the online survey platform Prolific on the basis of British nationality and residency in England and Wales, and successfully completed the survey. Of these, 187 (62%) were female and 113 (38%) were male; ages were from 18 to 72, *M* = 36.3, *SD* = 12.2; 16% were students; and 52% were in full-time employment, 21% in part-time employment, and 27% were unemployed or of other status. The study was calculated to have an 80% power to detect an effect size of *η*^2^ = 0.038. Participants were paid GBP 0.50 for their time. The study was conducted in accordance with approval obtained from UCL’s Research Ethics Committee (EP/2018/005). Informed consent was obtained from each individual in advance of participation by providing them with study information and a consent form to agree to.

#### 4.1.2. Design

We used a between-participant design with one independent variable with three levels in which all participants viewed the target apprentice scenario. Participants were randomly assigned to 3 conditions: a control condition where participants reviewed only the apprentice scenario; a replication condition where participants reviewed the captain scenario before the apprentice scenario; and a new test condition where participants additionally read closing submissions that highlighted the existence of an alternative solution for choosing who to be killed (namely, a lottery), before making their decision.

#### 4.1.3. Materials

All participants were advised at the outset that they would be asked to put themselves in the place of a juror in the Crown Court. Duress of circumstances was explained as a defence that was available where an accused commits what would otherwise be an offence in order to save a life or prevent serious harm to somebody. Participants were told that more details would be provided with the cases, and that they would be asked to decide one or two hypothetical cases where the accused argues duress of circumstances and asked to give a verdict.

All participants were asked to consider the apprentice scenario, which was in accordance with the materials used in Study 2. However, the legal issues were simplified slightly from that study. Participants were told that the prosecution accepted that the captain and mate may have believed that their lives were at risk, but argued that they were not acting reasonably by killing the apprentice. It was explained that because the burden of proof was on the prosecution: (1) if they were sure the accused were acting unreasonably by killing the apprentice, they should find them guilty; and (2) if they thought that the accused were or might have been acting reasonably by killing the apprentice, they should find them not guilty. Participants were asked to give a verdict, to assess the reasonableness of the accused’s actions, and to give reasons for their verdict.

For those participants assigned to the control condition, the apprentice scenario was the only scenario they were asked to assess. For participants assigned to the replication condition, they were asked to assess the captain scenario from Study 2 prior to the apprentice scenario. In the new test condition, participants viewed a variant of the apprentice scenario that included submissions by both the prosecution and the defence prior to making their decision. The prosecution’s submissions specifically drew attention to the argument that the behaviour of the accused was not reasonable because there was a better way to behave, namely, to randomly draw lots to decide who would be sacrificed.

After completing the relevant materials appropriate to each condition, participants completed an attention question in which they were asked to identify the issues from a selection of three: (1) whether the accused should be acquitted or found guilty, (2) how reasonable it was for the accused to kill when faced with starvation, and (3) whether the accused’s lives were at risk. Of these, the third was the erroneous answer given that participants were advised that the prosecution accepted that the accused may have believed that their lives were at risk. As before, participants were asked to indicate whether they found the description of the law easy or difficult to understand, and whether they had any previous legal experience.

#### 4.1.4. Measures

For each scenario, participants had to respond to the same measures. These were a binary indication of verdict, which was either guilty or not guilty; a gradated Likert response with 0.1 increments from 1 to 7 where each number was matched with a verbal description (1 = completely unreasonable; 2 = unreasonable; 3 = somewhat unreasonable; 4 = evenly balanced; 5 = somewhat reasonable; 6 = reasonable; 7 = completely reasonable); and a text response box to give an explanation of their decision. A Likert scale was preferred over the previous 0–100 percentage scale because of concerns that the percentage scale was insufficiently granular and sensitive, with the risk that participants focused primarily on the values of 0 or 100 with few meaningful gradations between these extremes. The 7-point Likert scale was also a more standardised measure and therefore would facilitate comparison with previous studies on moral and legal decision making.

Once participants had responded to one scenario or both, all participants responded to a series of other measures. An attention check invited the participants to identify the issues in the case with 2 correct answers (whether the accused should be acquitted or found guilty, how reasonable it was for the accused to kill when faced with starvation) and one incorrect answer (whether the accused’s lives were at risk). Participants were also asked if they found the explanation of the law easy or difficult, if they had any relevant legal knowledge or experience, and if they found any parts of the survey confusing or inconsistent.

#### 4.1.5. Procedure

Again, participants were recruited online from the Prolific platform and participated in the survey using the online platform Qualtrics in a place of their choosing, using their own device. On initial referral, participants were first given the study information form. They were then asked to complete the consent form comprising a number of statements to which an affirmative answer was required in order to participate in the survey. An anonymous user identification was collected to enable subsequent matching of demographic data without compromising the participants’ anonymity.

Participants were randomly assigned to one of the three conditions such that a third of the participants viewed the control condition, a third viewed the replication condition, and a third viewed the test condition. After viewing a scenario within a condition, they were first asked to give a verdict; second, asked to assess the reasonableness of the accused’s action on the Likert scale; and third, asked to explain their decision. Once they had completed one or two scenarios according to the condition, all participants were then posed the additional measures described above.

After completing the survey, participants were thanked for their participation and referred back to the Prolific survey platform to confirm their participation. Once both platforms had confirmed the participant’s successful completion of the survey, their remuneration was authorised.

### 4.2. Results

Three hundred participants completed the survey. In terms of the attention check, it was assessed that it may have been too challenging for participants, given that the ‘incorrect’ answer required participants to appreciate that the issue of whether the accused’s lives were at risk was only not a live issue because the prosecution in this version of the survey had conceded it. Nonetheless, many participants were sensitive to this, with only 9% of participants choosing this response. Relatedly, the explanation of the law in this survey seemed to be better understood by participants, with a lower percentage (9%) compared to Study 2 finding the explanation of the law difficult to understand. Again, only a small percentage (7%) had any legal experience, though for the most part, this was limited, and no reference was made to any familiarity with the case of *R v Dudley v Stephens*, and this would ordinarily not have been sufficient to exclude such participants from acting as jurors in such a case. As a result, the decision was taken not to exclude any participants.

Although the captain scenario was a manipulation rather than a condition, the incidental data from participants confirmed once again that this scenario was assessed as more favourable (reasonableness *M* = 3.93/7, *SD* = 1.71, verdicts guilty:not guilty = 42:56) than the apprentice scenario used in the control condition (reasonableness *M* = 3.05/7, *SD* = 1.68, verdicts guilty:not guilty = 72:30); see Figure 3, first panel. These differences were significant according to a *t*-test and a Fisher’s exact test, respectively (*t*(198) = 3.71, *p* < 0.001, 95% *CI* = [0.41, 1.36], Cohen’s *D* = 0.52; *OR* = 3.18, 95% *CI* = [1.71, 6.00]).

In terms of the hypotheses predicted for this study, the results again confirmed that the apprentice scenario in the control condition (Figure 3, second panel) was significantly more acceptable when reviewed in isolation than when it was reviewed after the captain scenario (Figure 3, first panel). A multiple linear regression was undertaken to assess the statistical significance of the difference with variables representing (1) the difference between the control condition of the apprentice scenario reviewed in isolation and the replication condition of the apprentice scenario reviewed after the captain scenario (Figure 3, third panel) and (2) the difference between the control condition of the apprentice scenario reviewed in isolation and the test condition of the apprentice scenario with submissions (Figure 3, fourth panel). The overall regression was statistically significant (*R*^2^ = 0.04, *F*(2,297) = 5.48, *p* < 0.01). As with Study 2, the variable representing the difference between the control condition and the replication condition was found to be statistically significant (*β* = −0.74, 95% *CI* = [−1.19, −0.30], *p* = 0.001, *η*^2^ = 0.35). However, while differences between the control condition and the new test condition were in the direction predicted (in that in the test condition, the accused’s behaviour was assessed as somewhat less reasonable), the corresponding variable was not statistically significant (*β* = −0.27, 95% *CI* = [−0.71, 0.17], *p* = 0.23, *η*^2^ = 0.005).

Predictably, an identical pattern was seen with participants’ verdicts, with participants much more likely to convict in the replication condition (guilty:not guilty = 90:8) compared to the control condition (guilty:not guilty = 72:30). A logistic regression was undertaken to assess the effect of the two manipulations on verdicts. The replication condition was again statistically significant (*OR* = 0.21, 95% *CI* = [0.08, 0.47], *p* < 0.001), whereas the new test condition (guilty:not guilty = 75:25) was not (*OR* = 0.80, 95% *CI* = [0.43, 1.49], *p* = 0.46).

### 4.3. Discussion

Study 3 replicated the effect previously seen in Study 2, whereby the generally unfavourably received apprentice scenario became even more unfavourably received when the generally more acceptable captain scenario preceded it. This order effect was in a different direction to that seen in Study 1 and previous research in the moral decision-making context. As such, the replicated effect is inconsistent with individual consistency-type theories that assume that the reason for this effect is that participants wish to appear consistent across the similar dilemmas, therefore altering their responses in subsequent dilemmas to be more akin to their responses in previous dilemmas. At the same time, the further results from the new test condition do not provide much support for explanations that assume that the order effects are due to the previous dilemmas drawing participants’ attention to salient information that the participants had not previously appreciated. The new test condition that included a prosecutor drawing participants’ attention to the information assumed to be salient (the fact that a lottery would have been a fairer way to select the individual to be killed and less likely to lead to the weakest sailor being taken advantage of) seemed to have very little effect on either assessments of reasonableness or verdicts, though the small effect that there was, was in the direction predicted. Thus, either salience explanations might not be appropriate, or the new manipulation might not have been as effective as the previous manipulation. Certainly, there are some differences between the two manipulations. The established manipulation consisted of participants actively making a decision on a gruesome scenario where a vulnerable crew member lost their life in extreme circumstances. By contrast, the new test manipulation amounted to a relatively brief, bald, and deliberately neutral statement. This leaves open the possibility that another manipulation that draws the information to the attention of participants in a more effective way might still achieve a similar effect.

Potentially linked to the strength of the manipulation is the issue of the validity of the paradigm, specifically the role of deliberation. As noted previously, commensurate with their seriousness, criminal cases of the type that form the basis of the vignettes in Studies 2 and 3 would invariably be prosecuted before the Crown Court. As such, these decisions would be arrived at by a group of jurors who seek to arrive at a consensus through deliberations that may take days or even weeks to complete. There is good empirical evidence that suggests that groups of decision makers are more effective than individual decision makers at identifying salient information ([17]; [22]; [41]; [52]; [53]; [80]). An ‘assembly bonus’ effect ([52]; [53]; [54]) often arises where the quality of group decision making is more than the sum of the individual contributions. To the extent that the types of order effects that we have disclosed are due to initial failures to take account of salient information, decision making in the more externally valid context of a jury might be more effective at identifying this salient information, thereby giving rise to the order effects identified by an alternative means.

## 5. Study 4

While the asymmetrical case order effects revealed in Studies 1, 2, and 3 provide equivocal evidence for salience-type explanations, this type of explanation remains the most consistent with the evidence seen. In Study 4, we therefore looked to take advantage of one of the characteristic aspects of the experimental paradigm that we had previously not focused on. As previously noted, the seriousness of the criminal dilemmas that we were posing to participants would mean that in the real world, they would be determined by a jury. In addition to decisions being made by laypeople, one of the other characteristics of jury decision making is that jurors arrive at a group decision through deliberation. To date, we had analysed decision makers as individuals, in common with much research into jury decision making ([19]; [34]; [36]; [44]; [63]). However, research suggests that group deliberation is more than simply an averaging mechanism between different views ([17]; [22]; [41]; [52]; [53]; [80]). Instead, there is evidence that, provided that there is both a diversity of views and the opportunity to debate, then group decision making can be superior to the sum of the equivalent number of individual decision makers ([54]; [80]). There is also evidence consistent with this effect in the context of legal decision making ([22]; [41]; [42]; [51]).

Given our speculation that the asymmetric order effects seen in the pairs of legal cases might be caused by one of the cases highlighting ideas that the participant had not previously realised as salient, we speculated that if this was the explanation, giving participants the opportunity to deliberate as a jury might provide a more effective means of causing the effects seen in Studies 2 and 3. Specifically, if the effect was caused by the captain scenario making participants realise that there was a means of deciding who to sacrifice that would not risk the weakest member of the group being killed at the expense of the strongest (i.e., a lottery), deliberation as a group might be more likely to reveal this idea to jurors than if they considered the dilemma in isolation. We therefore sought to see if we could replicate the effect using a condition whereby participants had an opportunity to deliberate as part of a group before assessing the unilateral scenario and a condition whereby participants assessed the same scenario purely as individuals.

If the asymmetrical order effect previously identified was caused by participants realising that there was a better means of selecting the sailor to be sacrificed than unilateral action by the more senior sailors, thereby casting the sailor’s behaviour in the apprentice scenario in a more negative light, then participants who had the opportunity to deliberate in groups might be more likely to identify this information than those required to take the decision by themselves, thereby giving rise to the effects seen in Studies 2 and 3.

### 5.1. Method

#### 5.1.1. Sample/Participants

One hundred and fourteen participants were recruited from students studying an undergraduate psychology methodology course at University College London. Of these, 85% were female and 15% were male. Ages were from 18 to 23 (*M* = 18.7, *SD* = 0.86). The study was calculated to have an 80% power to detect an effect size of *d* = 0.61. Participants participated as part of the methodology course and wrote up the experiment and results, but they were not informed of the theoretical background or predictions prior to participation. Participants were not financially remunerated for their participation. The study was conducted in accordance with approval obtained from UCL’s Research Ethics Committee (EP/2018/005). Informed consent was obtained from each individual in advance of participation by providing them with a study information sheet and a consent form to agree to.

#### 5.1.2. Design

We used a between-participant design with one independent variable with two levels in which all participants assessed the unilateral scenario previously used. Participants were randomly assigned to 2 conditions: an individual condition where participants assessed the scenario in isolation without the ability to discuss with others; and a group condition where participants assessed the scenario after having an opportunity to discuss the scenario with other members of a group. Whereas juries in England and Wales generally amount to 12 individuals, given the relatively small pool of available participants, it was decided to compromise at groups comprised of 6 individuals. In order to ensure a realistic prospect of an effect due to group membership, a greater overall proportion of students were assigned to the group condition so that a sufficiently large number of groups could be formed. A final ratio of 2:1 of individuals: groups (or 1:3 of participants as individuals: participants as group members) was chosen as a compromise designed to ensure both appropriate representation of participants as individuals and as group members, as well as a sufficiently large number of overall groups.

#### 5.1.3. Materials

Participants in all conditions were given written instructions advising them that they would be asked to put themselves in the place of a juror in a Crown Court to decide a hypothetical case. It was explained to them that, in law, an accused who would otherwise be convicted of an offence can avoid liability if they have a defence. The defence of duress of circumstances was described as where an accused commits what would otherwise be a crime in order to save a life or prevent serious harm to somebody. Participants were told that they would be presented with a case where the accused argued duress of circumstances and asked whether they thought the accused should be convicted or acquitted.

Participants were given a written document summarising the relevant facts of the scenario as described in Studies 2 and 3. Participants were told that the two were being prosecuted for murder but relied on the defence of duress of circumstances. This was described as where the accused committed the offence because they reasonably believed that they would die or be seriously injured, and a person of reasonable firmness would have acted in the same way.

All participants were asked individually to give a verdict of guilty or not guilty, to choose a reasonableness point on a 7-point Likert scale, and to give reasons for their decision. Participants in the group condition were additionally asked to complete the same measures by consensus after deliberating but before responding individually.

#### 5.1.4. Measures

All participants completed a single form which asked them to indicate a verdict of guilty or not guilty; reasonableness on the same 7-point Likert scale previously used (1 = completely unreasonable; 2 = unreasonable; 3 = somewhat unreasonable; 4 = evenly balanced; 5 = somewhat reasonable; 6 = reasonable; 7 = completely reasonable); and to explain the reasons for their decision. Participants were also separately asked demographic details, comprising their age and gender.

Participants in the group condition were additionally asked to complete the same form as a group by consensus after deliberating but before completing the same measures as an individual.

#### 5.1.5. Procedure

Participants were randomly assigned to either the individual or the group condition. Participants assigned to the group condition were randomly assigned further to a small group of 6. Those in the individual condition completed the study in an undergraduate laboratory in silence, supervised by university demonstrators. Those in the group condition participated together with their assigned group in private rooms. The demonstrators administered the survey, but they were excluded from the private rooms while the participants deliberated and completed the measures.

In the individual condition, participants were provided with an envelope containing the instructions, participant information sheet, consent form, materials, and measures. Participants were instructed to read the instructions and the participant information sheet and complete the consent form if they were happy to participate. They were then asked to read the facts of the case and complete the form giving their verdict, reasons, and demographic information. Participants were instructed to return all materials other than the participant information sheet. Demonstrators ensured that all materials were collected from the participants before they left.

Those participants in the group condition were additionally instructed to discuss the case, agree on a group verdict, and to provide reasons on behalf of the group before they completed the individual measures described above. Demonstrators similarly ensured that all materials were collected from the participants in the group condition before they left with the exception of the participant information sheet.

All participants participated in the study on the same afternoon. All individual participants began the study at the same time. Participants in the group condition were given staggered half-hour periods to attend the private rooms to participate.

### 5.2. Results

Of the one hundred and fifteen people who participated in the study, one individual was excluded on the basis that they were particularly familiar with the case of *R v Dudley v Stephens*; 29 people participated as individuals and 85 participated as members of a group. Those in the group condition were assigned to 15 groups of 6, but due to absences, some groups had fewer than 6 members. On the day, 11 groups comprised 6 members, with 3 groups of 5 members, and 1 group of 4 members.

In terms of the individual verdicts given by jurors who considered in isolation compared to those who deliberated as a member of a group, the proportions were uncannily similar. Those in the individual condition delivered verdicts in a ratio of 21:8 guilty:not guilty, a ratio of 0.72 when rounded, whereas those in the group condition delivered verdicts in a ratio of 61:24 guilty:not guilty, also a ratio of 0.72 when rounded. Obviously, the tiny difference between the two groups was not significant. A Fisher’s exact test confirmed this (*p* = 1, *OR* = 0.97, 95% *CI* = [0.33, 2.68]). There was somewhat more of a difference in individual reasonableness assessments between the conditions, but this was not in the direction that would be consistent with salience explanations. Individual reasonableness assessments in the individual condition rated the behaviour as slightly less reasonable (*M* = 3.55/7, *SD* = 1.62) than in the group condition (*M* = 3.81/7, *SD* = 1.43), see Figure 4. A Welch two-sample *t*-test indicated that this difference did not reach statistical significance (*t*(43.8) = 0.77, *p* = 0.45, Cohen’s *D* = 0.18). These reasonableness ratings were similar, but slightly higher, than participants’ responses to the equivalent scenario in Study 3, as might be expected given the younger nature of the Study 4 demographic.

An interesting, consistency-related, incidental finding of the study was that there was a strong correlation between verdicts given by the group and individual verdicts given by members of that group. That is, where a group arrived at a particular group verdict, almost all of the individual members were also likely to give that verdict when asked individually. Thus, of the 64 individuals who were in groups that collectively arrived at a verdict of guilty, only 7 subsequently gave an individual verdict of not guilty; and of the 21 individuals who were in groups that collectively arrived at a verdict of not guilty, only 4 subsequently gave a verdict of guilty. Unsurprisingly, a Fisher’s exact test confirmed a very strong correlation between group verdict and individual verdict (*p* < 0.001, *OR* = 32.1, 95% *CI* = [7.8, 171.7]). While some correlation would have been expected between the individual verdict and the group verdict (because the majority in a group would be assumed to have a greater influence on the final verdict), the correlation seemed to be much higher than would be expected. This suggested that either (1) individuals who would otherwise have preferred a different verdict had been persuaded to take a different position by the group discussions; and/or (2) that individuals who had agreed on a unanimous group verdict subsequently gave the same individual verdict to appear consistent, even if they would otherwise have preferred a different verdict.

### 5.3. Discussion

The findings of Study 4 implied little effect of group deliberation on individual participants’ decisions, but a deeper examination of the decision-making patterns suggests that there may also be more complicated factors at play. A statistical comparison of convictions and reasonableness ratings implied very little difference between the individual and group conditions. However, some care might need to be taken with this inference given there was evidence that other factors might have been involved. In particular, the distribution of individual verdicts within the groups showed a clear bimodal distribution with individual verdicts very strongly influenced by the eventual agreed-upon verdict of the group. Given that participants were randomly assigned to a particular group, there ought not to have been a bimodal distribution unless there were further processes happening. Often, individuals were also unanimously in agreement with the group verdict.

There seemed to be at least two possible explanations. One was an institutional-type consistency effect of the type discussed in the introduction ([45]). It is likely that the group deliberation caused opinions to be influenced in both directions prior to agreeing on a verdict ([12]; [17]). Once deliberation and consensus had been reached, it is conceivable that participants were at least partially motivated to give a verdict consistent with the majority verdict rather than the verdict that they would have preferred had they not been required to reach a group verdict by majority.

A second possibility was that there were other factors than the method of selection of the victims in the scenario that the group identified as salient and important, and which subsequently persuaded the group one way or another. From previous research, we know that jury, or group, decision making is more than the sum of its parts ([17]; [22]; [41]; [52]; [53]; [80]). We also know that the majority in jury decision making does not always prevail. In some cases, an argument from the minority succeeds in persuading the majority. Furthermore, while we had predicted a metaphorical ‘eureka’ moment when participants realised that there was a particular piece of information (a fairer way of selecting the victim that would be less likely to result in the death of the most vulnerable), the present scenario comprised a fairly complicated set of facts. Therefore, unlike other research where there is only one solution (e.g., [21]), the complex background matrix of facts might have given rise to a number of different pieces of information that may have been treated as relevant by the groups. Thus, the group verdict could have been influenced by more than one piece of information, and that information could have swayed the group in both directions.

The lack of a recording or transcript of the group deliberations compounded the difficulties in diagnosing what additional factors might have influenced the group deliberations. Had this information been available, a qualitative assessment of the information that influenced the debate may well have shed some light on the debates. An additional issue was the relatively small jury size necessitated by the limited sample size. Though most jury groups in the study consisted of six members, this is still half the size of the typical jury. It is difficult to know the extent to which the small group size influenced the nature of the debates and whether there is a minimum group size to achieve the discovery or ‘assembly bonus’ effects seen in group deliberation. Other factors relevant to the design include the specified time slots given to the juries, which may have exerted some pressure to arrive at a premature consensus. Real-life juries, by contrast, are not given a deadline, and deliberations may continue for a considerable time.

The unexpected influence of the groups is an interesting phenomenon worthy of further exploration, but it complicates the search for an explanation of the order effects seen in these paired dilemmas. Given the uncertain effect of the use of group deliberation as a manipulation, it is difficult to draw firm conclusions regarding the competing theses we are examining.

## 6. Study 5

For our final study, we sought to increase the strength of the more direct manipulation used in Study 3. Whereas the information about the selection process assumed to be salient was communicated plainly in Study 3, in Study 5, we communicated this information more colourfully to ensure it was comprehended by participants, and also directly questioned participants following their participation to confirm that the information had been successfully communicated. The same criminal justice paradigm was chosen, and the information was conveyed using a memorable illustration from sailing history. This was the history of the wreck of the American whaling ship Essex ([15]). In 1820, the ship was sunk by a Sperm Whale while 2500 miles off the coast of South America. After the ship was destroyed, the crew put to sea in small whaleboats, which then became separated. The sailors on the boat with the captain, George Pollard, survived for 2 months at sea before they ran out of food and began to die. Initially, the surviving seamen cannibalised the bodies. Once the bodies of the seamen who had died of natural causes had been consumed, the crew decided to draw lots to decide who would be killed for the survival of the others. Owen Coffin, the captain’s 17-year-old cousin, drew the black spot. The captain had sworn to protect Coffin and offered to protect him, but Coffin reportedly said ‘No, I like my lot as well as any other.’ Further lots were then drawn to decide who would be the one to kill Coffin. Coffin was killed, and the others consumed his body. Two sailors eventually survived to tell the story. If salience theories best explain the asymmetric order effects that our research has revealed, participants in the condition where the prosecutor highlights the option of a fairer means of selecting the sailor to be killed would be expected to assess the behaviour of the sailors as less reasonable and correspondingly be more likely to convict.

### 6.1. Method

#### 6.1.1. Sample/Participants

Two hundred participants were recruited using the online survey platform Prolific on the basis of British nationality and residency in England and Wales, and successfully completed the survey. Of these, 130 (65%) were female and 70 (35%) were male; ages were from 18 to 71, *M* = 37.3, *SD* = 12.1; 15% were students; and 53% were in full-time employment, 17% in part-time employment, and 30% were unemployed or of other status. The study was calculated to have an 80% power to detect an effect size of *d* = 0.40. Participants were paid GBP 0.60 for their time. The study was conducted in accordance with approval obtained from UCL’s Research Ethics Committee (EP/2018/005). Informed consent was obtained from each individual in advance of participation by providing them with study information and a consent form to agree to.

#### 6.1.2. Design

We used a between-participant design with one independent variable with two levels in which all participants viewed the target apprentice scenario. Participants were randomly assigned to the 2 conditions: a control condition where after reviewing the scenario, participants read neutral submissions by the prosecution that essentially described the facts and the law; and an experimental condition where after reviewing the scenario, participants read much more colourful submissions by the prosecution, referring to the facts of the shipwreck of the Essex and how the survivors had fairly selected who to kill using a lottery.

#### 6.1.3. Case Materials

Participants in both conditions were given written instructions advising them that they would be asked to put themselves in the place of a juror in a Crown Court to decide a hypothetical case. It was explained to them that, in law, an accused would otherwise be convicted of an offence can avoid liability if they have a defence. The defence of duress of circumstances was described as where an accused commits what would otherwise be a crime in order to save a life or prevent serious harm to somebody. Participants were told that they would be presented with a case where the accused argued duress of circumstances and asked whether they thought the accused should be convicted or acquitted.

Participants read a document summarising the relevant facts of the unilateral scenario as described in Studies 2, 3, and 4. To reprise, this was that a small 3-man crew had been shipwrecked on a lifeboat without communication equipment and had exhausted their rations over a period of 2 weeks. The captain and mate secretly discussed the situation and decided that they might survive for a little longer if they resorted to cannibalism. They agreed to kill the apprentice who they felt would die soonest. After the apprentice fell asleep, the mate held him down while the captain killed him. They subsequently consumed him. Participants were told that the two were being prosecuted for murder, but they were relying on the defence of duress of circumstances. This was described as where the accused committed the offence because they reasonably believed that they would die or be seriously injured, and a person of reasonable firmness would have acted in the same way.

In the control condition, the prosecution’s closing speech was neutral and descriptive, summarising the facts and the law. In this speech, the prosecution simply asserted that the actions of the accused were not reasonable and therefore they should be convicted. In the experimental condition, the prosecution’s closing speech also asserted that the actions of the accused were not reasonable, but supported this assertion by reference to the facts of the case of the whaling ship Essex. The submissions explained that in that case, the crew had decided to draw lots, and when a junior member of the crew, Coffin, was selected, the captain had offered to protect him, but Coffin had refused to allow him. Participants were told that the crew then drew lots again to decide who would execute him.

All participants then read the same defence submissions and the judge’s summary of the law, where the law, including the burden of proof, was explained. Participants were advised in the summary of the law that the only issue was whether what the accused did was reasonable, and if they thought that the accused’s actions were, or might have been, reasonable, they should find them not guilty, and if they were sure that the accused’s actions were not reasonable, they should find them guilty.

All participants were asked individually to give a verdict of guilty or not guilty, to choose a reasonableness point on a 7-point Likert scale, and to give reasons for their decision as before. Additionally, at the conclusion of the study, participants were subsequently asked to explain what (if anything) the accused could have done differently when faced with these circumstances. Participants were asked to identify the issues in the case, to state how easy they found the explanation of the law, and whether they had any relevant legal experience or knowledge.

#### 6.1.4. Measures

All participants responded to the same measures. These were a binary indication of verdict which was either guilty or not guilty; a gradated Likert response with 0.1 increments from 1 to 7 where each number was matched with a verbal description (1 = completely unreasonable; 2 = unreasonable; 3 = somewhat unreasonable; 4 = evenly balanced; 5 = somewhat reasonable; 6 = reasonable; 7 = completely reasonable); and a text response box to give an explanation of their decision.

After making their decision, participants were asked what the accused might have done differently (if anything) when faced with this situation and provided with an open text response box to explain.

Once participants had responded to one scenario or both, all participants responded to a series of other measures. An attention check invited the participants to identify the issues in the case with two correct answers (whether the accused should be acquitted or found guilty, and how reasonable it was for the accused to kill when faced with starvation) and one incorrect answer (whether the accused’s lives were at risk). Participants were also asked if they found the explanation of the law easy or difficult, and if they had any relevant legal knowledge or experience.

#### 6.1.5. Procedure

As with Study 3, participants were recruited online from the Prolific platform and participated in the survey using the online platform Qualtrics in a place of their choosing, using their own device. On initial referral, participants were first given the study information form. They were then asked to complete the consent form comprising a number of statements to which an affirmative answer was required in order to participate in the survey. An anonymous user identification was collected to enable subsequent matching of demographic data without compromising the participants’ anonymity.

Participants were randomly assigned to one of the two conditions by the Qualtrics platform in a balanced way, so that half were in the control condition with the neutral prosecution submissions, and half were in the experimental condition with the prosecution submissions explaining about the shipwreck of the Essex and the procedure adopted by the shipwrecked sailors during that event. After viewing the scenario, and the prosecution and defence submissions, participants were first asked to give a verdict, secondly to assess the reasonableness of the accused’s actions, and thirdly to give reasons for their decision.

Once participants had completed their responses to the scenario, they were then asked what the accused might have done differently and then the additional measures.

After completing the survey, participants were thanked for their participation and referred back to the Prolific survey platform to confirm their participation. Once both platforms had confirmed the participant’s successful completion of the survey, their remuneration was authorised.

### 6.2. Results

Two hundred participants successfully completed the survey. Of these, 84% found the law easy to understand, with only 16% finding it difficult. In terms of issues identified, only 1% of participants exclusively selected the legally incorrect issue of whether there was a risk to the life of the accused, and 8% selected this in combination with one or more of the other issues. This suggested a generally good understanding of the law, as, though the issue of whether there was a risk to the lives of the accused was a condition for the availability of the defence of duress of circumstances, it was not an issue on the facts presented because the prosecution had conceded that there was a risk to the accused’s lives. Seven percent of the participants had some legal training or experience, but this was universally limited and did not suggest any familiarity with the precedents referred to in the materials. As such, no exclusions were made.

In terms of the planned contrasts, there was little appreciable difference between the conditions, see Figure 5. Those in the experimental condition, Essex, were slightly more likely to convict (68%) than in the control condition (63%). However, while in the direction predicted, this small difference did not reach statistical significance according to a Fisher’s exact test (*OR* = 0.79, 95% *CI* = [0.42, 1.47]). Likewise, participants assessed the accused’s behaviour in the experimental Essex condition as slightly less reasonable (3.03/7) than in the control condition (3.09/7), but this modest difference did not approach statistical significance on a *t*-test (*t*(198) = 0.25, *p* = 0.80, 95% *CI* = [−0.40, 0.52], Cohen’s *D* = 0.04).

Notwithstanding the lack of an apparent difference between the conditions, it seemed that participants in the Essex experimental condition were much more likely to be aware of the alternative of adopting a fairer means of choosing the victim compared to those in the control condition. The open text responses to the question about what the accused might have done differently were coded by a coder blind to the experimental conditions. Those participants who either referred to adopting a form of lottery or to discussing the situation to reach a consensus were coded as having identified the possibility of a fairer selection process. Other responses that were coded as not having identified this possibility included responses such as that the accused should have waited longer, sought other means of sustenance, or simply refrained from cannibalism. Only 9% of those in the control condition were coded as having identified the possibility of a fairer means of selection compared to 37% of the experimental condition. This difference was statistically significant pursuant to a Fisher’s exact test (*OR* = 5.82, 95% *CI* = [2.54, 14.68], *p* < 0.001), indicating the manipulation in the experimental condition had successfully communicated the information to a significantly higher proportion of participants in that condition.

### 6.3. Discussion

Study 5 addressed the possibility that the manipulation used in Study 3 was not strong enough. The information about a fairer means of selection was conveyed in a more direct way, confirmed by the analysis, which indicated that a much larger proportion of participants were aware of the option to use a lottery in the experimental condition. Nonetheless, the information appeared to make little or no difference to participants’ responses between the conditions. Salience-type theories would assume that the asymmetrical order effects revealed in Studies 1, 2, and 3 are due to previous scenarios drawing participants’ attention to the option of a fairer means of selection, an option that many may have otherwise overlooked. As such, the explicit communication of this information ought to have had a similar effect, but it did not.

It remains possible, if unlikely, that the manipulation used in Studies 3 and 5 was not sufficiently strong to cause the same effects seen in Study 2 and replicated in Study 3. In Study 5, a considerably larger portion of participants in the experimental condition (approximately two-fifths) disclosed this as an idea when asked, compared to a smaller proportion in the control condition (approximately one-tenth). Still, this may not have been a sufficiently large proportion to lead to a detectable effect size. Equally, the sailor who was killed following the shipwreck of the Essex was also a junior sailor, information which may not have been enough to trigger the salience of a lottery as a means to prevent more vulnerable victims being exploited. Against this, group deliberation also did not appear to have an appreciable effect in Study 4, notwithstanding the more complicated underlying effects of group deliberation. Overall, Study 5 provides little support for salience-type explanations to account for the asymmetrical order effects identified in our research.

## 7. General Discussion

The primary goal of our research was to replicate the case order effects seen in moral decision making where cases are determined sequentially in a legal context. The studies undertaken demonstrated emphatically that the same phenomenon occurs, both in civil and criminal law contexts (Studies 1, 2, and 3). The research also went beyond the order effects seen in the moral decision-making domain because the legal decision-making order effects we have demonstrated manifest in both directions: some responses to cases determined second become both more similar to the cases that precede them (Study 1), whereas other cases become more dissimilar (Studies 2 and 3). Regardless of whether the effect was in one direction or the other, consistent with moral decision-making research, we also found that these order effects were asymmetrical, with responses in one of the two cases being labile when determined second, with responses to the other case remaining stable (Studies 1, 2, and 3). The implication that legal decision makers will determine some cases differently depending on the order that they are presented is concerning, on the face of it, as it suggests that outcomes may be somewhat arbitrary. The challenge is particularly pertinent in the legal context, given adherence to the doctrine of stare decisis. Stare decisis encourages decision makers determining the law to follow previous decision makers, even if they might otherwise determine the case differently. As such, the view taken by initial decision makers can be very influential in determining the course of the law, with their influence being amplified by later decision makers. Addressing arbitrary effects of case order may be particularly problematic in the real world because the order in which cases come before the courts is essentially random and therefore very difficult to foresee or influence, in contrast to the controlled conditions characteristic of experimental research. How concerning these effects are depends on their theoretical explanation. It would be worrying if these effects were caused by motives that do not serve the ends of the legal system. By contrast, if they are caused by decision makers realising in later decisions that they had earlier overlooked salient information, meaning that later decisions are more comprehensive, then the patterns might not be so problematic. To the extent that the phenomenon identified is problematic, addressing it in the real world may be challenging due to the lack of institutional control over case order. Policy approaches might include relaxing the rigidity of the principle of stare decisis where there are limited prior precedents. However, the appropriateness of policy recommendations is heavily dependent on establishing a firm theoretical foundation.

Given these considerations, the secondary goal of our research was to shed light on why this phenomenon occurs. Two main explanations for these effects have been propounded, and while these explanations might be relevant to some effects, we found limited support for these to explain our research findings. Perhaps the most common explanation, that participants strive to maintain an appearance of individual consistency between the paired scenarios, is only compatible with Study 1. Furthermore, the asymmetrical nature of the effects in Studies 1, 2, and 3 implies that additional assumptions are required in addition to a simple desire to appear consistent. For example, some have argued that some scenarios are so unambiguous that it leaves little scope for participants to change their response to appear consistent with their earlier responses, which might explain the stability of responses ([11]; [20]; [66]; [73]). However, individual consistency is difficult to square with the findings in Studies 2 and 3 because participant responses in scenarios presented second became more dissimilar to their responses to preceding scenarios. The asymmetrical and bidirectional nature of the effects in studies 1, 2, and 3 would, on the face of it, be more compatible with salience-type theories because responses in the scenario that is the source of the salient information would be expected to be stable, whereas responses to the target scenario, where responses to the salient information would necessarily be labile. However, Studies 3, 4, and 5 provide limited support for salience-type theories because, if salience explanations are correct, it would be reasonable to assume that it would also be possible to trigger a similar pattern by explicitly communicating the salient information to participants. Studies 3, 4, and 5 appear to belie this, though: disclosing the relevant information in a neutral way (Study 3); in a more lively way (Study 5); or facilitating the discovery of the assumed salient information by allowing group deliberation (Study 4) appeared to have a negligible effect on responses. Again, it is possible to posit other auxiliary hypotheses, such as stochastic factors, the greater effectiveness of actively using information to make a decision compared to passively receiving the information, or other countervailing effects, but at present, these are speculative. It would be invaluable to have some theory to enable reliable predictions as to when these effects will occur, which requires further work.

When undertaking further work, some consideration needs to be given to methodology. In order to replicate effects previously identified in moral decision making, the design of our Studies 1, 2, and 3 adopted that used in trolley research of reversing the order of paired dilemmas. However, in the trolley context, this was originally simply a means to account for any unforeseen effects not of direct relevance to the hypotheses being tested. As we have seen, the asymmetrical nature of the findings implies that simply reversing the order will not always account for unexpected effects, an observation relevant to psychology methodology generally. More specifically for our purposes, this design is not ideal for isolating individual effects because randomising the order gives rise to more than two different conditions. For example, with two dilemmas, A and B, presented in a random order, there are a number of different conditions: A in isolation; B in isolation: A after B; and B after A. The result is that this limited design makes it less straightforward to identify the causes of the identified effects, so a more sophisticated experimental design would be advisable.

Against this background, the scope for further research is fairly wide. The replication of the phenomenon identified with professional adjudicators is always valuable, notwithstanding the known similarities between lay and professional decision making ([10]; [66]; [75]; [78]). However, professional adjudicators may behave differently where they are drawing on a body of specialist knowledge. If something like salience theory accounts for the order effects seen, this could differentiate lay and professional adjudication on the basis that the latter might be less likely to overlook salient information. However, given the cost and difficulty of such research and the limited insight that we currently have, the priority might be to first gain a deeper understanding of the phenomenon we have identified with lay participants.

Equally, replication in other jurisdictions might also be helpful. The experiments conducted as part of this research were based on the law of England and Wales due to the background of the researchers and the pool of available participants. However, legal scenarios tested (murder, assisting suicide) are not peculiar to English and Welsh law, so, absent cultural differences, there might be fewer reasons to expect different results in other jurisdictions. Additionally, the use of lay participants unfamiliar with legal norms specific to a jurisdiction provides some assurance of the generalisability of these results.

## Figures and Tables

**Figure 1 behavsci-15-01250-f001:**
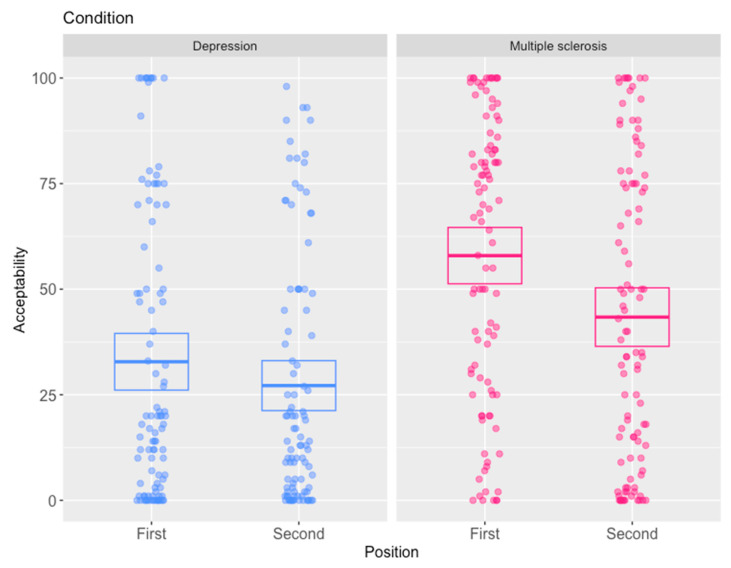
Mean acceptability ratings by condition and position. Boxes show mean and 95% *CI*.

**Figure 2 behavsci-15-01250-f002:**
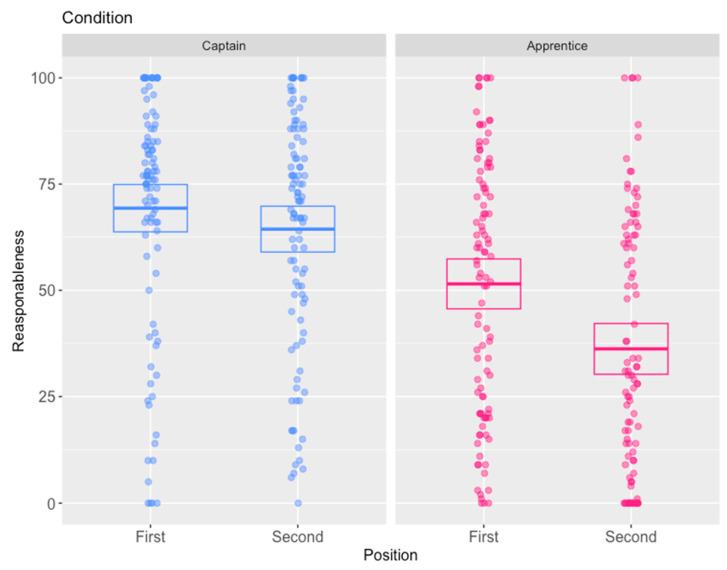
Mean reasonableness assessments by condition and position. Boxes show mean and 95% *CI*.

**Figure 3 behavsci-15-01250-f003:**
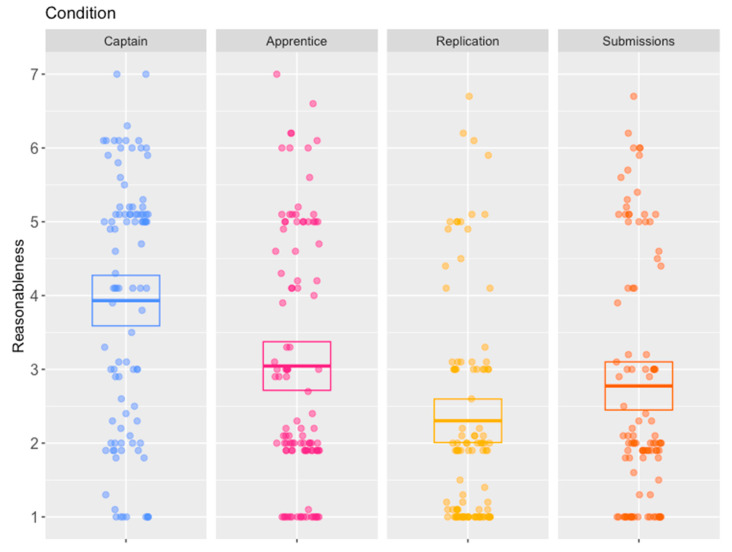
Mean reasonableness ratings of one of the manipulations (first panel) and of the three conditions (second, third, and fourth panels). Boxes show mean and 95% *CI*.

**Figure 4 behavsci-15-01250-f004:**
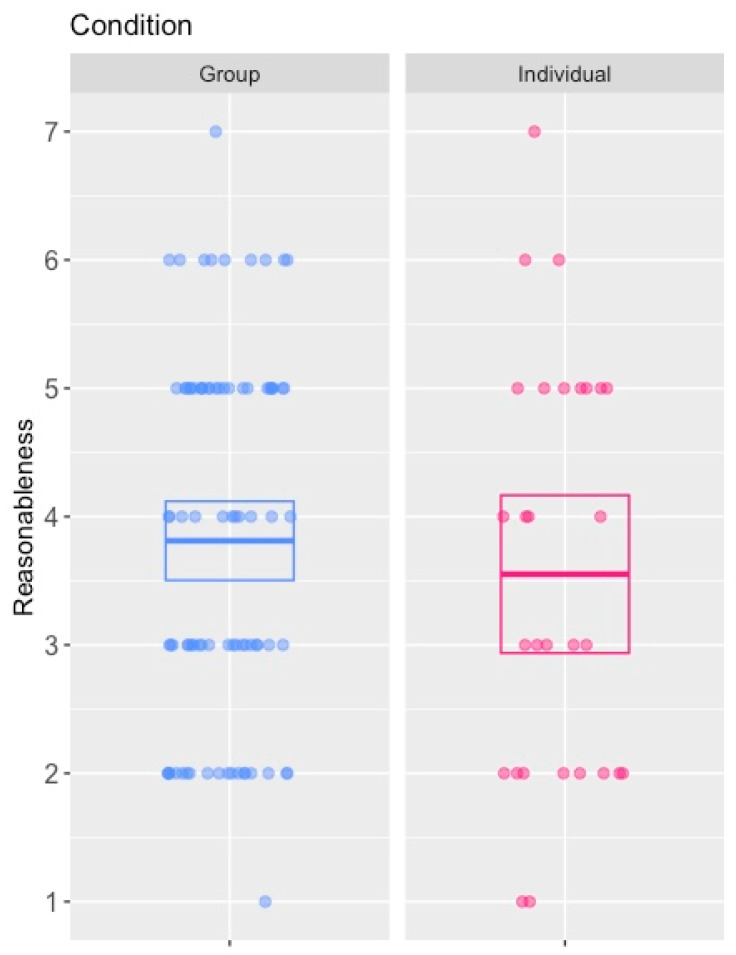
Mean reasonableness ratings by jurors deliberating as a group prior to decision, compared to those deciding as individuals in isolation. Boxes show mean and 95% *CI*.

**Figure 5 behavsci-15-01250-f005:**
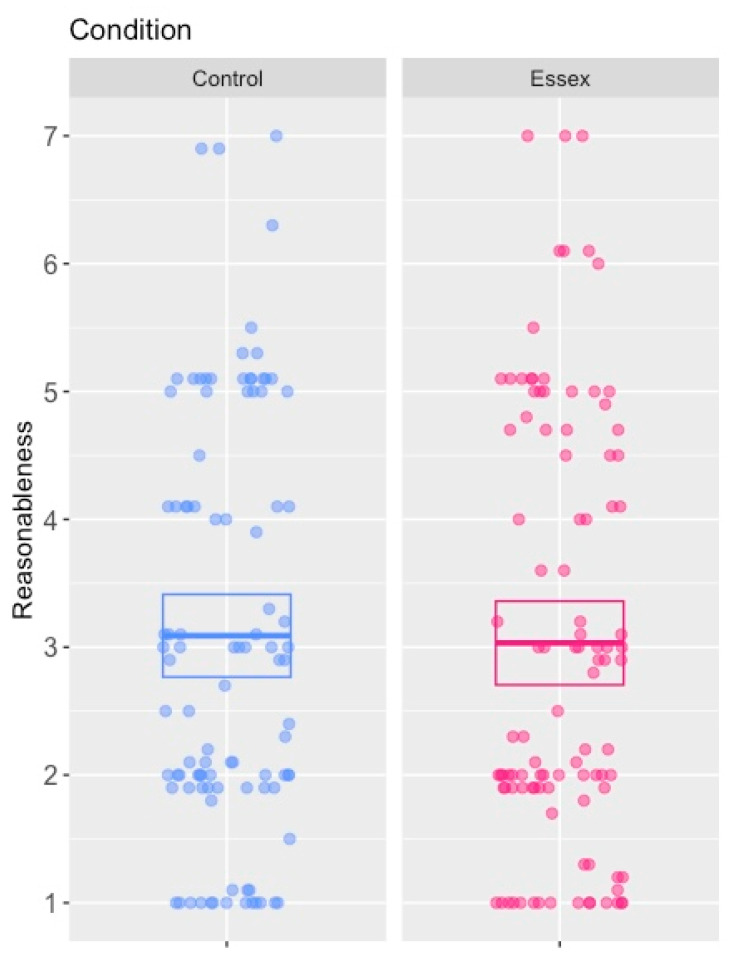
Mean reasonableness assessments comparing the control condition where participants were given no information with the condition where participants were advised of the facts of the shipwreck of the whaler Essex. Boxes show mean and 95% *CI*.

## Data Availability

The raw data supporting the conclusions of this article will be made available by the authors on request.

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
