# Peer review of "Case Order Effects in Legal Decision-Making"

_behavsci, 2025, doi:10.3390/bs15091250_

Round 1
Reviewer 1 Report
Comments and Suggestions for Authors
Thank you for the opportunity to review this manuscript. Overall, this is a solid and well-executed study that makes a valuable contribution to the literature on legal decision-making. However, I would like to raise a few concerns for your consideration.
Major Points
1. Legal Implications
The study demonstrates that order effects are bidirectional and that the underlying mechanism remains unclear. However, it is not entirely clear what legal implications can be drawn from these findings. The manuscript mentions the principle of stare decisis, but in actual legal practice, it is impossible to randomize the order of binding decisions as one might in a psychological experiment. Therefore, it is difficult to see how the current findings regarding order effects translate into institutional or procedural implications.
Please consider adding a clearer explanation of the potential legal implications of these findings. Alternatively, if the lack of mechanism identification limits such implications, it may be appropriate to explicitly acknowledge that the current study offers limited practical implications and that further research is needed to clarify this.
2. Generalizability
The scenarios appear to be based on English law, and the participant sample is restricted to English-speaking individuals. This raises potential concerns about the generalizability of the findings to other legal systems and cultural contexts. I recommend that you explicitly discuss these limitations in the manuscript.
Minor Points
3. Classification of the Civil Case in Study 1
Study 1 references the Suicide Act in creating the scenario, and the case is described as a "civil case." However, assisting suicide is a criminal offense under the Act. Could you clarify why this was classified as a civil case, or revise this terminology to reflect the correct legal context?
4. Basis for Effect Size in Power Calculation
In Study 1, the sample size calculation is based on an assumed effect size of d = .40. Could you explain the rationale for selecting this effect size? Was it based on prior research or a theoretical expectation?
Author Response
Major Points COMMENT 1. Legal Implications The study demonstrates that order effects are bidirectional and that the underlying mechanism remains unclear. However, it is not entirely clear what legal implications can be drawn from these findings. The manuscript mentions the principle of stare decisis, but in actual legal practice, it is impossible to randomize the order of binding decisions as one might in a psychological experiment. Therefore, it is difficult to see how the current findings regarding order effects translate into institutional or procedural implications. Please consider adding a clearer explanation of the potential legal implications of these findings. Alternatively, if the lack of mechanism identification limits such implications, it may be appropriate to explicitly acknowledge that the current study offers limited practical implications and that further research is needed to clarify this. RESPONSE 1: It is a pertinent point that phenomena that can be identified under controlled experimental conditions are much harder to identify (or influence) in the real world. This is one reason why experiments are used instead of passive observation. It is therefore correct to note that a phenomenon that is harder to demonstrate in the real world is also harder to influence in the real world. In the present circumstances, the lack of a firm theoretical explanation also makes it hard to give clear institutional recommendations. These issues are already noted in the text, but further comments have been added in Section 7. COMMENT 2. Generalizability The scenarios appear to be based on English law, and the participant sample is restricted to English-speaking individuals. This raises potential concerns about the generalizability of the findings to other legal systems and cultural contexts. I recommend that you explicitly discuss these limitations in the manuscript. RESPONSE 2: This is an insightful point. However, against this, the scenarios tested (murder, suicide) are not specific to English law and are likely to be similar in other jurisdictions. Additionally, the fact that subjects were lay participants insulated them to a degree from knowledge about any peculiar aspects of English and Welsh law that might exist. A comment to this effect is included in section 7. Minor Points COMMENT 3. Classification of the Civil Case in Study 1 Study 1 references the Suicide Act in creating the scenario, and the case is described as a "civil case." However, assisting suicide is a criminal offense under the Act. Could you clarify why this was classified as a civil case, or revise this terminology to reflect the correct legal context? RESPONSE 3: While assisting suicide is a criminal offence, if a party wishes to seek legal guidance in advance as to whether proposed behaviour is criminal or not, this is sought from a civil court. This is explained at section 2, which has been clarified slightly. COMMENT 4. Basis for Effect Size in Power Calculation In Study 1, the sample size calculation is based on an assumed effect size of d = .40. Could you explain the rationale for selecting this effect size? Was it based on prior research or a theoretical expectation? RESPONSE 4: The D value chosen equated to a 'medium' effect size, chosen due to the lack of equivalent previous legal research. This is clarified in section 2.1.1.Reviewer 2 Report
Comments and Suggestions for Authors
The purpose of this manuscript was to examine whether case order, frequently found in moral decision making, are also present in the legal decision-making context. If so, to determine the underlying reason why they occur (due to individuals’ desire to be consistent vs. the salience of certain information contained in one scenario compared to the other). Results revealed evidence of asymmetrical order effects such that responses to some scenarios remained stable regardless of whether they were presented first or second, whereas responses to the other scenario changed as a function of the order they were presented (first vs. second).
Overall, this manuscript offers important contributions to the literature and has several strengths. First, it addresses a novel topic (case order effects in legal decision making) with limited prior research, but with important implications for judicial decision making. Second, the introduction is grounded in theory on moral decision making that supports the importance of the research and provides relevant background. Third, the methodology is strong—the manuscript consists of 5 separate studies with adequate power, several of the studies pre-tested the different scenarios to ensure their validity, the materials were adapted from previously established stimuli in moral psychology but with added ecological validity, and Study 4 utilized a jury deliberation paradigm which is often lacking in the field of juror decision making.
I do, however, have some suggestions and comments for the authors to consider:
- On p. 9 (lines 368-379), the authors do a nice job discussing how judges’ decisions might differ from laypeople’s and what the implications are. Another point the authors may consider noting, if they’d like, is that even though laypeople (rather than judges) are making these decisions, a parallel can be drawn to the morality research from which the scenarios are drawn. For example, in the Transplant dilemma, laypeople are asked to take the role of the surgeon. Thus, it is not out of the ordinary (especially for this particular area of study) to ask participants to take the role of a professional and make a decision in an area that they normally wouldn’t do in their own lives.
- I suggest the authors move the sentences at lines 187-200 (which provide a summary of the results) to the discussion section rather than before they even get into the methods of each study.
- There are some places throughout the manuscript where redundancy can be reduced. For example, some of the methods sections (e.g., procedures, measures) both within and across studies can be streamlined by referring readers to the previous study and only highlighting differences that exist.
- The researchers collected open-ended data across their studies, I’m curious if they conducted any analyses on these data (apart from those noted in Study 5)? (If not, that’s not problematic, this point is just out of curiosity.)
- Study 4: One participant was excluded due to their familiarity with the case of R v Dudley v Stephens; however, in Study 2, 1% of participants were familiar with the case, yet none were not excluded. Why were different exclusion criteria used?
- If neither individual consistency nor salience clearly account for the pattern of results, what else might explain them? I applaud the researchers’ use of different methods and materials to try to pinpoint the salience type explanation for their findings; however, perhaps it’s not only about the salience of informational content (i.e., that fairer selection criteria could be used). Perhaps it’s about the negative emotionality that’s evoked in the apprentice scenario, and how this emotionality is exacerbated to an even greater extent when subsequently contrasted against the captain scenario? That is, the captain scenario (which is more acceptable, sympathetic, fairer) creates a contrast effect that exacerbates disapproval, moral outrage, and moral disgust at the apprentice scenario to an even greater degree than when the apprentice scenario is read in isolation.
- (Although Study 5 more closely approximates the emotional angle via incorporation of the Essex story in the prosecutor’s statements, this is still a little different than participants actively making two separate decisions (one for the captain condition and a second, subsequent one for the apprentice condition), and the contrast effects that might be elicited.)
- The authors might consider some of the legal decision-making research regarding emotionally evocative evidence (e.g., gruesome photographs), the roles of moral disgust and moral outrage, and how they relate to the need/desire to punish. This works illustrates that even when the evidentiary/informational value is the same (i.e., the exact same photo), photos presented in color (vs. black-and-white) elicit greater outrage and disgust and in turn greater punishment. A parallel might be drawn for the captain vs. apprentice scenario. In addition, people are more likely to experience moral outrage when vulnerable or innocent individuals are the ones who are harmed (Darley & Pittman, 2003), with moral judgments being most severe when there is a greater degree of victim suffering and greater intent to cause harm on the part of the perpetrator (Gray et al., 2012). These elements also characterize the apprentice scenario.
- Darley, J. M., & Pittman, T. S. (2003). The psychology of compensatory and retributive justice. Personality and Social Psychology Review, 7(4), 324–336. https://doi.org/10.1207/S15327957PSPR0704_05
- Gray, K., Young, L., & Waytz, A. (2012). Mind perception is the essence of morality. Psychological Inquiry, 23(2), 101–124. https://doi.org/10.1080/1047840X.2012.651387
- Phalen, H. J., Salerno, J. M., & Nadler, J. (2021). Emotional evidence in court. In Research handbook on law and emotion(pp. 288-311). Edward Elgar Publishing.
- Salerno, J. M. (2017). Seeing red: Disgust reactions to gruesome photographs in color (but not in black and white) increase convictions. Psychology, Public Policy, and Law, 23(3), 336.
- This rationale would align with some of what the authors already note. For example, when they state, “Certainly, there are some differences between the two manipulations. The established manipulation consisted of participants actively making a decision on a gruesome scenario where a vulnerable crew member lost their life in extreme circumstances. By contrast, the new test manipulation amounted to a relatively brief, bald, and deliberately neutral, statement” (lines 830-834).
- Minor point: I think there is supposed to be a “not” inserted in this sentence to say, “not guilty”: “Thus, of the 64 individuals who were in groups that collectively arrived at a verdict of guilty, only 7 subsequently gave an individual verdict of guilty…” (lines 1001-1002).
Author Response
COMMENT 1: On p. 9 (lines 368-379), the authors do a nice job discussing how judges’ decisions might differ from laypeople’s and what the implications are. Another point the authors may consider noting, if they’d like, is that even though laypeople (rather than judges) are making these decisions, a parallel can be drawn to the morality research from which the scenarios are drawn. For example, in the Transplant dilemma, laypeople are asked to take the role of the surgeon. Thus, it is not out of the ordinary (especially for this particular area of study) to ask participants to take the role of a professional and make a decision in an area that they normally wouldn’t do in their own lives. RESPONSE 1: Good point. Text to this effect added at s.2.3. COMMENT 2: I suggest the authors move the sentences at lines 187-200 (which provide a summary of the results) to the discussion section rather than before they even get into the methods of each study. RESPONSE 2: This is a style choice. For comprehensibility, we prefer to provide a short summary of the findings before the experiments. COMMENT 3: There are some places throughout the manuscript where redundancy can be reduced. For example, some of the methods sections (e.g., procedures, measures) both within and across studies can be streamlined by referring readers to the previous study and only highlighting differences that exist. RESPONSE 3: Some similarity between the experiments has already been removed, but given the number of differences between experiments 2-5 we feel there isn't much more that could be reduced without introducing some confusion or the necessity to repeatedly refer back to previous experiments. COMMENT 4: The researchers collected open-ended data across their studies, I’m curious if they conducted any analyses on these data (apart from those noted in Study 5)? (If not, that’s not problematic, this point is just out of curiosity.) RESPONSE 4: No, open-ended data such as qualitative responses were not analysed further. COMMENT 5: Study 4: One participant was excluded due to their familiarity with the case of R v Dudley v Stephens; however, in Study 2, 1% of participants were familiar with the case, yet none were not excluded. Why were different exclusion criteria used? RESPONSE 5: The different choices were because in Study 2, no participants had much legal experience or familiarity with R v D & S and decided in isolation so it was unlikely their responses would be influenced. By contrast in Study 5, the individual excluded indicated a good familiarity with the case and they may have been in a group where they might have influenced others, so the decision was made to exclude them. A comment is made in the text. COMMENT 6: If neither individual consistency nor salience clearly account for the pattern of results, what else might explain them? I applaud the researchers’ use of different methods and materials to try to pinpoint the salience type explanation for their findings; however, perhaps it’s not only about the salience of informational content (i.e., that fairer selection criteria could be used). Perhaps it’s about the **negative emotionality** that’s evoked in the apprentice scenario, and how this emotionality is exacerbated to an even greater extent when subsequently contrasted against the captain scenario? That is, the captain scenario (which is more acceptable, sympathetic, fairer) creates a contrast effect that exacerbates disapproval, moral outrage, and moral disgust at the apprentice scenario to an even greater degree than when the apprentice scenario is read in isolation. - (Although Study 5 more closely approximates the emotional angle via incorporation of the Essex story in the prosecutor’s statements, this is still a little different than participants actively making two separate decisions (one for the captain condition and a second, subsequent one for the apprentice condition), and the contrast effects that might be elicited.) - The authors might consider some of the legal decision-making research regarding emotionally evocative evidence (e.g., gruesome photographs), the roles of moral disgust and moral outrage, and how they relate to the need/desire to punish. This works illustrates that even when the evidentiary/informational value is the same (i.e., the exact same photo), photos presented in color (vs. black-and-white) elicit greater outrage and disgust and in turn greater punishment. A parallel might be drawn for the captain vs. apprentice scenario. In addition, people are more likely to experience moral outrage when vulnerable or innocent individuals are the ones who are harmed (Darley & Pittman, 2003), with moral judgments being most severe when there is a greater degree of victim suffering and greater intent to cause harm on the part of the perpetrator (Gray et al., 2012). These elements also characterize the apprentice scenario. Darley, J. M., & Pittman, T. S. (2003). The psychology of compensatory and retributive justice. Personality and Social Psychology Review, 7(4), 324–336. https://doi.org/10.1207/S15327957PSPR0704_05 Gray, K., Young, L., & Waytz, A. (2012). Mind perception is the essence of morality. Psychological Inquiry, 23(2), 101–124. https://doi.org/10.1080/1047840X.2012.651387 Phalen, H. J., Salerno, J. M., & Nadler, J. (2021). Emotional evidence in court. In Research handbook on law and emotion(pp. 288-311). Edward Elgar Publishing. Salerno, J. M. (2017). Seeing red: Disgust reactions to gruesome photographs in color (but not in black and white) increase convictions. Psychology, Public Policy, and Law, 23(3), 336. This rationale would align with some of what the authors already note. For example, when they state, “Certainly, there are some differences between the two manipulations. The established manipulation consisted of participants actively making a decision on a gruesome scenario where a vulnerable crew member lost their life in extreme circumstances. By contrast, the new test manipulation amounted to a relatively brief, bald, and deliberately neutral, statement” (lines 830-834). RESPONSE 6: This is a very interesting thought and might well form the basis of a new theoretical perspective on why these order effects occur. However, it may well need more development. For example, it is not entirely clear from a theoretical point of view why if a scenario is assessed as more emotionally outrageous when it follows a less emotionally outrageous scenario compared to when it is seen in isolation. Without some sort of theoretically underpinning explaining this link, there is a risk that this falls into redescription of the phenomena rather than a freestanding theory. Given this need for further philosophical development, we believe that this sort of theorising would be better carried out in a separate paper. COMMENT 7 Minor point: I think there is supposed to be a “not” inserted in this sentence to say, “not guilty”: “Thus, of the 64 individuals who were in groups that collectively arrived at a verdict of guilty, only 7 subsequently gave an individual verdict of guilty…” (lines 1001-1002). RESPONSE 7: Agree, change made.